# Kinetics of mRNA nuclear export regulate innate immune response gene expression

Diane Lefaudeux[1,2], Supriya Sen[1,3], Kevin Jiang[1,2], Alexander Hoffmann [1,2,3] ✉ & the UCLA Ribonomics Group*

The abundance and stimulus-responsiveness of mature mRNA is thought to be determined by nuclear synthesis, processing, and cytoplasmic decay. However, the rate and efficiency of moving mRNA to the cytoplasm almost certainly contributes, but has rarely been measured. Here, we investigated mRNA export rates for innate immune genes. We generated high spatio-temporal resolution RNA-seq data from endotoxin-stimulated macrophages and parameterized a mathematical model to infer kinetic parameters with confidence intervals. We find that the effective chromatin-to-cytoplasm export rate is gene-specific, varying 100-fold: for some genes, less than 5% of synthesized transcripts arrive in the cytoplasm as mature mRNAs, while others show high export efficiency. Interestingly, effective export rates do not determine temporal gene responsiveness, but complement the wide range of mRNA decay rates; this ensures similar abundances of short- and long-lived mRNAs, which form successive innate immune response expression waves.

Gene expression is key to cellular identity and function. Its regulation is complex allowing for gene-specific control of the abundance of gene products and the speed at which their abundances may be adapted to changes or perturbations. Considering mRNA, the gene expression intermediate whose cytoplasmic presence allows for protein synthesis, its cytoplasmic abundance is a function of numerous molecular reactions that are grouped within the broader terms of nuclear mRNA synthesis (transcription initiation, elongation), mRNA processing and export (e.g., capping, splicing, polyadenylation, and transport), and cytoplasmic degradation. Studies of immune response gene expression have been insightful as the dynamical nature of immune gene expression reveals the underlying kinetics of the regulatory steps. In response to pathogen-derived substances such as endotoxin, the expression of hundreds of genes is rapidly induced[1]. While the primary regulatory steps control nuclear mRNA synthesis[2], mRNAs show widely different cytoplasmic half-lives (ranging from just a few minutes to many hours), which thereby contributes to the responsiveness of gene expression[2,3].

What remains less well characterized is to what extent mRNA processing and export regulate gene expression. Genetic perturbation of the splicing machinery can diminish the abundance of mature mRNAs[4] and incompletely spliced mRNAs may be degraded via the nuclear exosome[5–7]. Indeed, mRNA export is mediated by RNA-binding proteins that are recruited to exon–exon junction complexes (EJCs). Recent studies have shown that while 3′-end cleavage and polyadenylation are always rapid, many genes have one intron that is spliced post-transcriptionally, potentially introducing delays in the appearance of cytoplasmic mature mRNA[8,9]. However, the resulting effective transport rates have not been measured quantitatively and it remains unknown to what extent these rates may be gene-specific or whether they contribute to the regulation of gene expression.

Here, we leveraged the high inducibility of innate immune gene expression programs to measure effective mRNA chromatin-to-cytoplasmic transport rate ("export rate") associated with each immune response gene. We produced genome-wide mRNA measurements in chromatin, nucleoplasmic, and cytoplasmic compartments at high temporal resolution, and developed a mathematical modeling workflow to infer kinetic rate constants and their associated confidence intervals. We report that the mRNA export rates vary over a 100-fold range among genes, but surprisingly do not contribute much

---

[1]Department of Microbiology, Immunology, and Molecular Genetics, University of California, Los Angeles, CA 90095, USA. [2]Institute for Quantitative and Computational Biosciences, University of California, Los Angeles, CA 90095, USA. [3]Molecular Biology Institute, University of California, Los Angeles, CA 90095, USA. *A list of authors and their affiliations appears at the end of the paper. ✉e-mail: ahoffmann@ucla.edu

to the temporal responsiveness of immune response gene expression, which is primarily controlled by cytoplasmic mRNA half-life. Instead, export rates determine the efficiency of transport (in the face of nucleoplasmic decay) and show a high correlation with cytoplasmic mRNA-degradation rates. Thereby, highly responsive genes with short half-lives are expressed highly thanks to highly efficient transport, and later waves of immune responsive genes with long half-lives are not disproportionately overexpressed due to lower efficiency transport.

## Results

### A detailed, quality dataset of endotoxin-induced mRNA synthesis and transport

To study the kinetics of post-transcriptional mRNA transport and decay of immune response genes we developed an experimental approach to follow mRNA expression within the cell. When mRNA is being transcribed it is linked to the chromatin-bound polymerase and may be isolated as chromatin-associated RNA (caRNA). It is then released by 3'-end cleavage and polyadenylation into the nucleoplasm (npRNA) and exported to the cytoplasm (cytoRNA) (Fig. 1A). We produced high spatio-temporal resolution RNA-seq data of three biological replicates (see Supplementary Fig. S1 for reproducibility of the replicates) by deeply sequencing RNA from three subcellular fractions (chromatin-associated, nucleoplasmic, and cytoplasmic) prepared

from murine bone-marrow-derived macrophages (BMDMs) at twelve timepoints within 2 h of stimulation with the endotoxin analog Lipid A, (Fig. 1B, see "Methods"). As observed for the *Tnf* gene, intronic reads are still present in the caRNA samples, but less in the npRNA samples and transcripts are fully spliced in the cytoplasmic samples (Fig. 1C). To enable a reliable quantification of mRNA expression for downstream analysis we selected strongly inducible genes (based on caRNA data, Fig. 1D). For the 273 selected genes, the npRNA expression profile is more similar to the caRNA expression profile than the cytoRNA expression profile (Fig. 1E). Correlation analysis shows that, overall, npRNA only slightly lags behind caRNA but that cytoRNA expression is less well correlated and more delayed (Fig. 1F).

To avoid bias in the caRNAseq data due to partially transcribed mRNA, genes were quantified based on the exonic portion of their last 5 kb. This required highly accurate annotation of the dominant transcription end site (TES). Thus, every selected gene track coverage was checked against GENCODE annotation database and 62 discrepant genes were removed, leaving 211 for further analysis (see "Methods", Supplementary Fig. S2A–F, and Supplementary Data 1). The observed TESs were largely consistent (within +/−100 bp) with an established database of polyA sites[10] based on 3'-end sequencing data (Supplementary Fig. S2G, right panel). For some genes (bottom right of Supplementary Fig. S2G, right panel) the observed TESs are even more

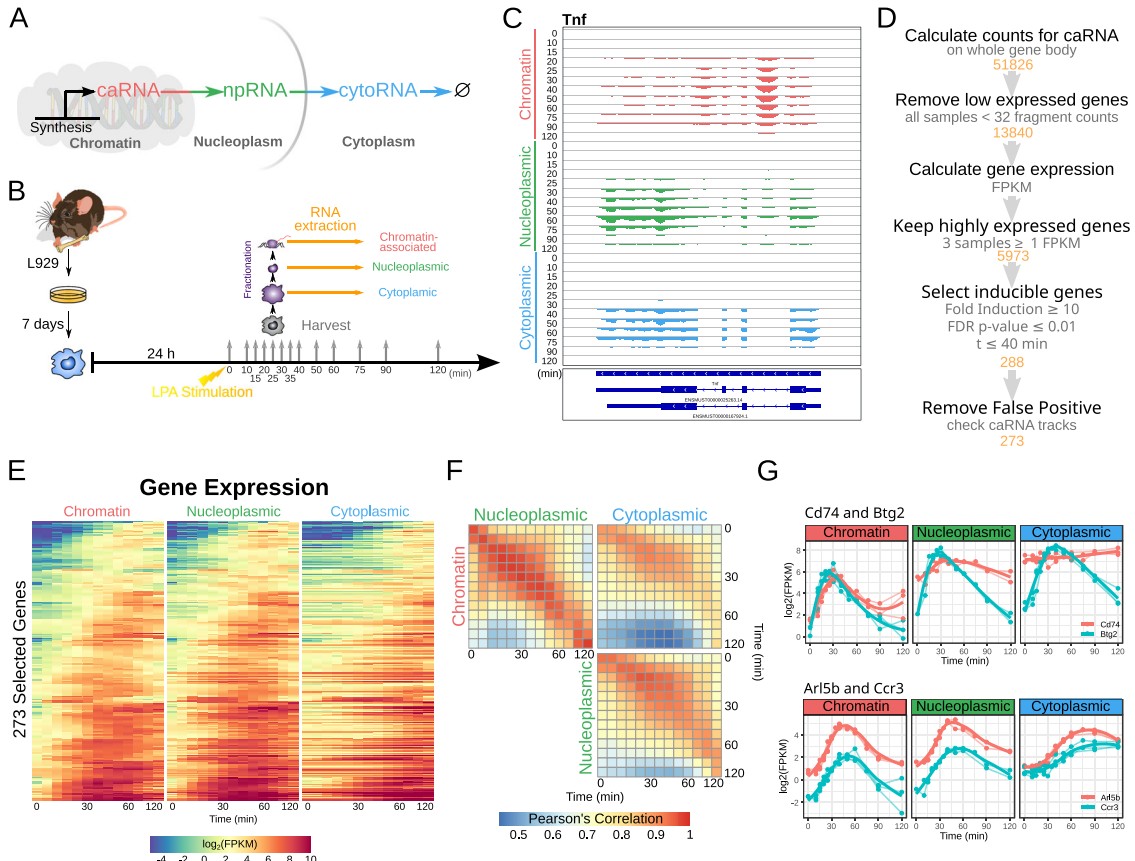

**Fig. 1 | Determining the kinetics of post-transcriptional events. A** Following mRNA from transcription to degradation. First, the mRNA is attached to the chromatin (caRNA), then it is released to the nucleoplasm (npRNA) upon 3'-cleavage, finally it is transported to the cytoplasm where it is degraded. **B** Schematic of the experimental setup. Bone-marrow-derived macrophages (BMDMs) are stimulated with LPA. Cells are harvested at different time after stimulation and fractionated into subcellular fractions, and RNA is extracted. **C** Example of tracks for the *Tnf* gene. Many intronic reads are found in the chromatin-associated fraction, fewer in the nucleoplasmic fraction, and the cytoplasmic fraction RNA is fully spliced. **D** Gene selection workflow. Lowly expressed genes were filtered out, and inducible genes at

the chromatin level were selected. **E** Heatmap of gene expression of the selected genes in (**D**) for each fraction. **F** Correlation analysis of mRNA abundance in each fraction. Note a stronger correlation between chromatin and nucleoplasmic fraction for nearby timepoints than between nucleoplasmic and cytoplasmic fractions or between cytoplasmic and chromatin fractions. Also, note a time shift, that timepoints on the chromatin correlate better with later timepoints in the cytoplasmic fraction. **G** Example genes: Top, *Cd74* and *Btg2* have similar chromatin expression profiles but exhibit different profiles in the nucleoplasmic and cytoplasmic fraction. Bottom, *Arl5b* and *Ccr3* show similar cytoplasmic levels but Arl5b expression is few fold higher than *Ccr3* in both chromatin and nucleoplasmic fractions.

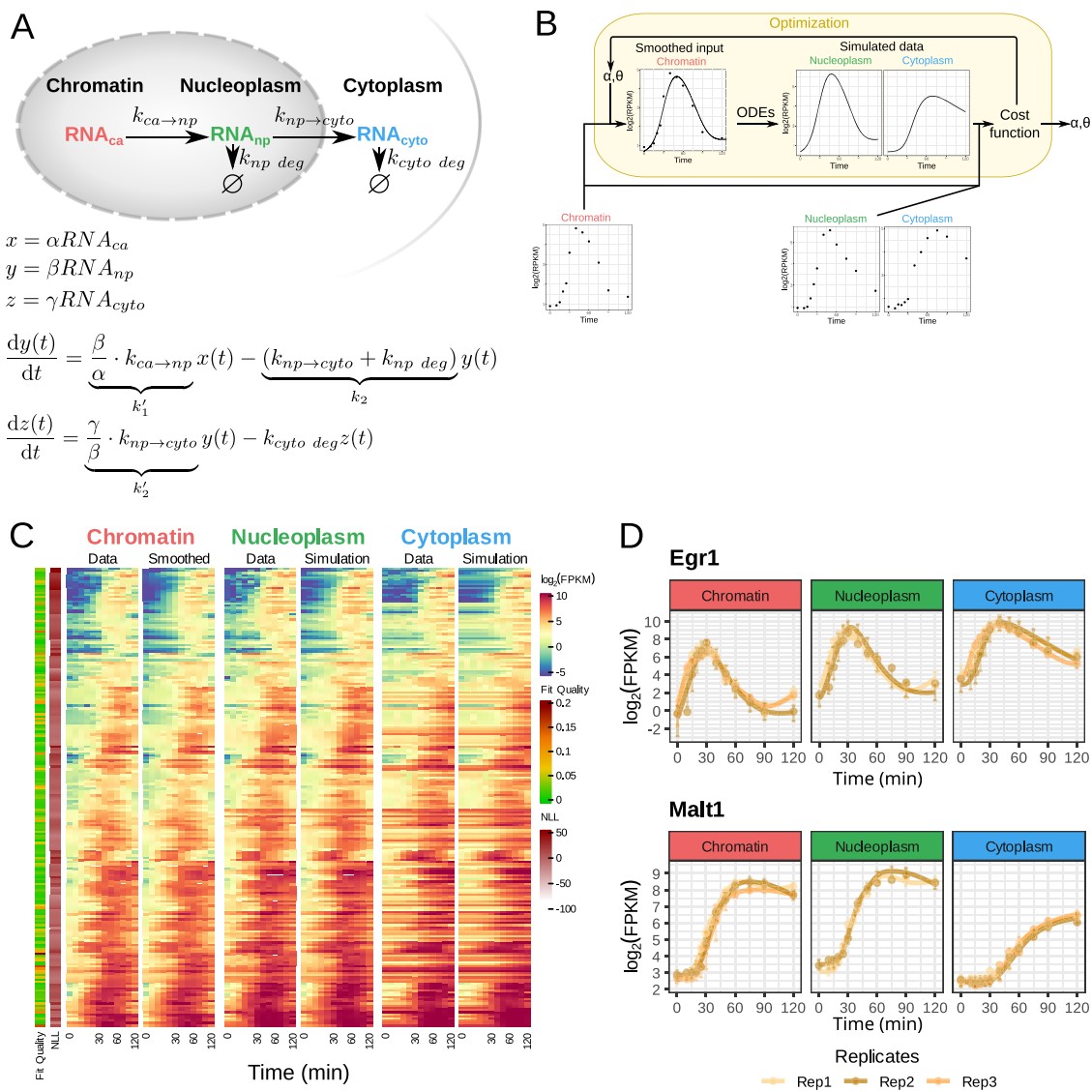

**Fig. 2 | Fitting a kinetic model to the stimulus-response data. A** Model linking mRNA measurements in different subcellular fractions, with associated equations. **B** Schematic of the fitting workflow that takes the expression of caRNA as input, simulates the nucleoplasmic and cytoplasmic mRNA abundances, and iterates to identify the optimal parameter set for each gene. **C** Heatmap of fitting results alongside the data. Most genes fit the data really well, as indicated in the Fit Quality bar on the left-hand side. Source data are provided as a Source Data file. **D** Example of fitting results for *Egr1* and *Malt1* genes for the three replicates. The error bars represent the 95% prediction interval of the model given the data library size.

consistent with the database than the information provided in the gene annotation (discrepant by >500 bp).

Tracking RNA expression in time and space illustrates that some genes, such as *Cd74* and *Btg2*, that have very similar caRNA expression, may exhibit very different npRNA or cytoRNA expression profiles (Fig. 1G, top panel); or that some genes with different caRNA expression profiles exhibit very similar cytoRNA expression profiles, such as *Arl5b* and *Ccr3* (Fig. 1G, bottom panel). This suggests that kinetic parameters of post-transcriptional processes may be gene-specific and may regulate gene expression dynamics.

## A mathematical model of mRNA dynamics to derive kinetic transport and decay rates

We developed a simple mechanistic mathematical model (Fig. 2A) to simulate the abundance of mRNAs in the different subcellular fractions. The model uses the measured caRNA expression profile as input to calculate npRNA and cytoRNA abundances over time as a function of kinetic parameters describing transport and decay. Given that RNA sequencing allows for relative quantification across genes and

samples, the parameter values that may be derived by fitting the model to the data are also relative. These relative parameters are denoted: $k_1'$ for the fractional appearance rate of mRNA in the nucleoplasm (in npFPKM/caFPKM min$^{-1}$); $k_2$ for the mRNA disappearance rate from the nucleoplasm (in min$^{-1}$), determined by both nucleoplasmic decay and nucleoplasm-to-cytoplasm transport; $k_2'$ for the fractional appearance rate of mRNA in the cytoplasm (in cytoFPKM/npFPKM min$^{-1}$), termed "the nucleoplasm-to-cytoplasm transport rate"; and $k_{cyto\ deg}$ for the cytoplasmic decay rate (in min$^{-1}$).

We fit the model to the expression data for each gene to estimate these kinetic parameters using an optimization pipeline with a cost function defined by the negative log-likelihood of reproducing the experimental data given the model; for the error model, we used a negative binomial distribution to account for both biological variability and sampling error for lowly expressed timepoints (schematized in Fig. 2B and described in "Methods"). A visual comparison of model-simulated and measured data graphed for all genes in a heatmap (Fig. 2C) or line graphs for two sample genes (Fig. 2D) illustrate the quality of the fits (see Supplementary Document 1 for detailed graphs

of individual genes). As the negative log-likelihood depends on the expression level, we developed a "fit quality" metric that also includes autocorrelation of the residuals (see Methods). Most genes exhibit excellent fits (Supplementary Fig. S3A) with fit quality values <0.06 (e.g., *Tnfaip2* and *Il10* have fit quality scores of 0.011/0.014 and 0.050/ 0.041, for the two replicate datasets, respectively, Supplementary Fig. S3B, top row). Only 9 genes have scores ≥0.06, and 2 have scores ≥0.1 for both replicates (e.g., *Cpd* and *Cd44* have fit quality scores of 0.096/0.084 and 0.15/0.012, respectively, Supplementary Fig. S3B, bottom row). Poor fit quality is typically due to discrepancies with the data from the nucleoplasmic fraction.

## Model fitting reveals which kinetic rate constants are identifiable from the data

Using the profile likelihood method, we also computed the 95% confidence interval of the estimated parameters (Supplementary Data 2). While $k_2'$ and $k_{cyto\_deg}$ were identifiable for almost every gene ($k_2'$: 207 and 202 for replicate 1 and 2, respectively; $k_{cyto\_deg}$: 199 for both replicates out of the 211 fitted genes), $k_1'$ and $k_2$ were identifiable for only ~130 genes ($k_1'$: 142 and 117; $k_2$: 144 and 116 for replicate 1 and 2, respectively; Fig. 3A left panel, see "Methods" for confidence interval estimation for each individual gene). However, in most cases where these parameters were not fully identifiable the lower bound could be found ($k_1'$: 46/69 and 50/95; $k_2$: 44/67 and 49/94 for replicate 1 and 2, respectively). Moreover, the confidence interval of the identifiable parameters was relatively narrow, around threefold between the upper bound and lower bound (i.e., +/− 1.8×) for $k_2'$ and $k_{cyto\_deg}$, and about tenfold (i.e., +/− 3.2×) for $k_1'$ and $k_2$ (Fig. 3A, right panel). To compare genes, we first examined the distribution of parameters that were identifiable; their values spread over a broad range, around 100-fold for $k_2'$ and $k_{cyto\_deg}$, and about 30-fold for $k_1'$ and $k_2$ (Fig. 3B). The parameter distributions resulting from fitting each replicate separately were similar (with a Kolmogorov–Smirnov distance of <0.14 between replicates for all parameters). Examining each parameter individually revealed that optimally fitted values (for genes that yielded identifiable parameter values) were reproducible across replicates (Fig. 3C), with the estimated values differing by less than +/− 2× for most genes (red dashed lines). Even when $k_1'$ and $k_2$ parameters were not identifiable, their ratio ($k_1'/k_2$), was almost always identifiable (202 and 198 for replicates 1 and 2 respectively) and highly reproducible (Fig. 3D), and this ratio's spread across the tested genes was much narrower than the individual parameters (only threefold).

To obtain an external validation, we compared the model-inferred mRNA half-life values to estimates obtained using Actinomycin-D treatment (Supplementary Fig. 4). The two methods led to very similar results (Fig. 3E) with a spearman rank correlation of 0.8 (*P* value $<2 \times 10^{-16}$). Interestingly, model inference led to a broader range of estimated half-lives than estimates with actinomycin-D experiments. The actinomycin-D approach estimated half-lives of within 15–300 min, whilst model-inferred half-lives ranged from 1 to over 1000 min. This reflects the shortcomings of actinomycin-D experiments in estimating very short and long half-lives[11].

## Nuclear export efficiencies and effective transport rates are highly gene-specific

One meaningful composite variable, $k_2'/k_2$, represents the efficiency of the nuclear export, i.e., how much mRNA arrives in the cytoplasm versus how much leaves the nucleoplasm by either export or nucleoplasmic decay. This measure spreads over 30-fold ($10^{1.5}$, Fig. 4A, left panel) meaning that if we assume no loss for the most efficiently transported genes, then for the least efficiently transported genes only ~3% of the nucleoplasmic mRNA will actually arrive in the cytoplasm. Similar to $k_2$, the export efficiency was not always identifiable (Fig. 4A, right panel), but we were able to identify it with the present dataset for ~130 genes out of the 211 (144 and 118 for replicates 1 and 2,

respectively) fitted genes, and for most unidentifiable genes the 95% confidence upper bound could still be defined (43/67 and 49/93 for replicates 1 and 2, respectively). For the genes for which this composite variable was identifiable the estimated value was also highly reproducible (Fig. 4A, middle panel).

Another composite variable is $k_1'k_2'/k_2$ which describes how fast a gene's caRNA transcript reaches the cytoplasmic fraction. We denote this composite measure "effective transport rate", and it is composed of the fractional cytoplasmic appearance rate $k_1'$ multiplied the nuclear export efficiency $k_2'/k_2$. This composite variable spreads across an even wider range of values: 100-fold between the fastest genes and the slowest (Fig. 4B, left panel). This effective transport rate is highly reproducible (Fig. 4B, middle panel) and almost always identifiable (203 and 198 for replicate 1 and 2, respectively, out of the 211 fitted genes; Fig. 4B, right panel).

These composite measures, "export efficiency" and "effective transport rate", correlate strongly (Fig. 4C, Pearson's correlation of -0.82; 0.83 and 0.81 for replicates 1 and 2, respectively) for the genes for which they are identifiable. The genes for which the export efficiency was not identifiable tend to have a lower effective transport rate. Given that the effective transport rate is identifiable for almost all genes and is strongly correlated with export efficiency, we focused on this composite parameter in subsequent analyses.

Interestingly, cytokines and chemokines have relatively high effective transport rates (Fig. 4D), as do some inflammatory transcription factors (*Junb*, *Egr1/2*, *Fos*, *Fosb*), while for others it is lower (*Fosl2*, *Irf1*, *Rel*, *Relb*). Negative feedback genes span a broad range in effective transport rates, with negative regulators of MAPK having higher values than negative regulators of NFκB. Genes involved in cell growth and cell adhesion tend to locate at the lower to medium range of the effective transport distribution.

Examining two genes on opposite ends of the effective transport rate range (*Egr1* and *Malt1*), we observe that even though they both reach similar levels on the chromatin, *Egr1*, which has a higher effective transport rate, is present at higher levels in the cytoplasmic fraction than Malt1 (Fig. 4E). This is the case even though *Malt1* has a longer cytoplasmic half-life than *Egr1* (~135 min for *Malt1* and ~15 min for *Egr1*).

## Transport parameters correlate with gene structure and sequence motifs rather than epigenetic signatures

To determine if the effective transport rate is an intrinsic gene characteristic or if it is context-dependent, we examined gene structure characteristics. We found that the effective transport rate is significantly anti-correlated with gene length (Fig. 5A) and the number of introns (Fig. 5B). Short genes with few introns have higher effective transport rates (potentially mediated by TPR) than longer ones with more introns, whose transport may depend on exon–exon junction complexes. Interestingly, even though cytokines and chemokines have similar gene lengths and intron numbers, cytokines tend to have higher effective transport rates than chemokines.

To examine if the need to splice in the nucleoplasm (rather than on the chromatin) might slow the effective transport rate, we calculated the percentage of spliced junction reads over the total of junction reads (spliced or unspliced) for each intron in the nucleoplasmic fraction using SIRI[12] and assumed that each intron is independently spliced. Interestingly, this measure of post-transcriptional splicing in the nucleoplasm, correlates with the effective transport rate even more strongly (Fig. 5C), though not necessarily due to a single bottleneck intron (Supplementary Fig. S5A). These data suggest that when splicing is not completed co-transcriptionally, it slows the effective mRNA transport rate to the cytoplasm.

To examine if RNA-binding proteins (RBPs) may also play a role in regulating the effective transport rate, we tested for enrichment of number of motifs for known RBP in the 5′-UTR and 3′-UTR of the genes and tested against the estimated effective transport rate

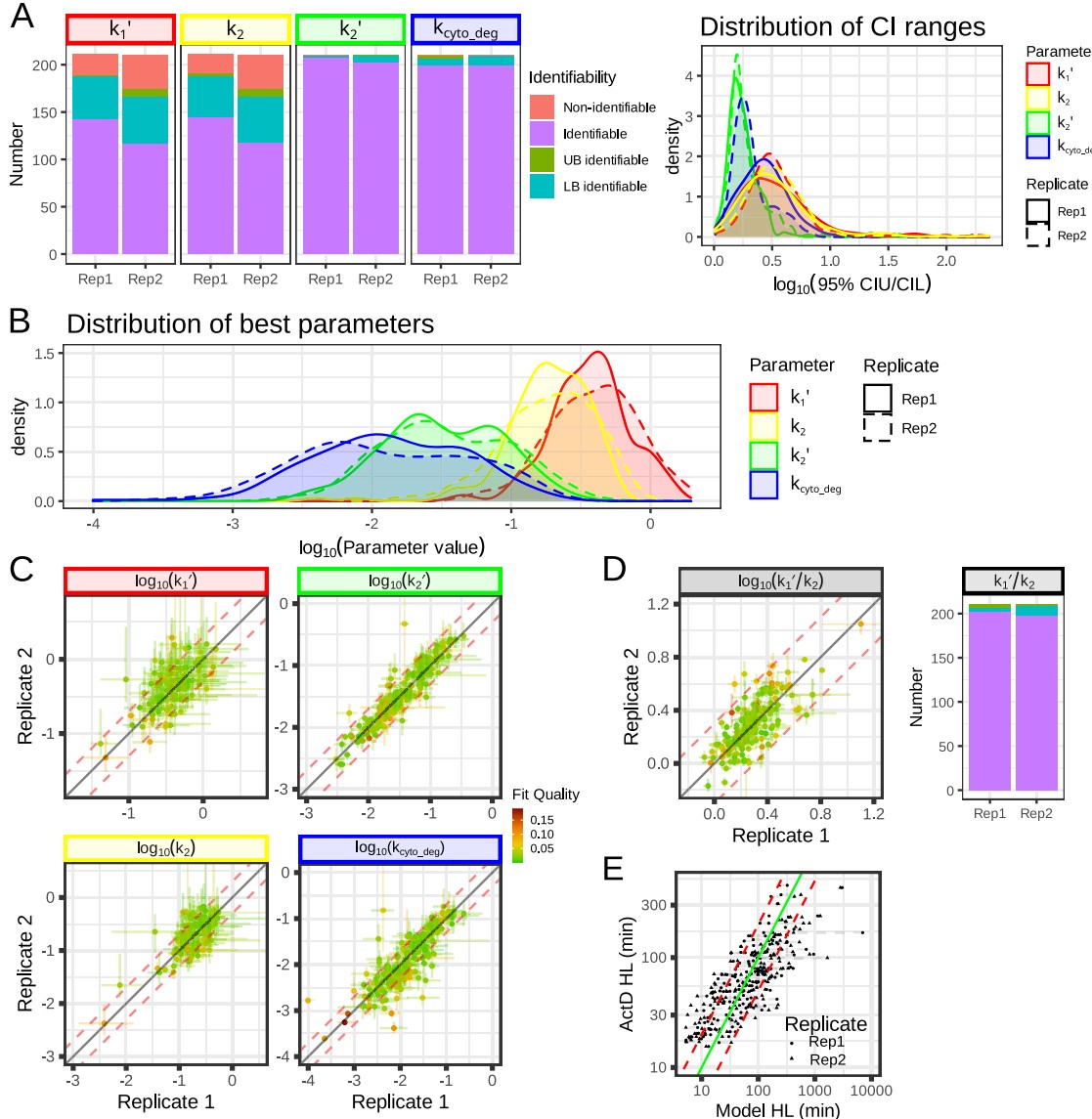

**Fig. 3 | Parameter inference, identifiability, and reproducibility. A** Number of genes for which parameters were identifiable, partially identifiable, or non-identifiable (left panel). Distribution of 95% confidence interval range for identifiable parameters. Also see the Supplementary Data file. Source data are provided as a Source Data file. **B** Distribution of the parameter values for the different replicates. The distributions are similar for the different replicates and the bulk of the distributions span a 30 to 100-fold difference between the genes depending on the parameter. **C** Reproducibility of each parameter. The color of the point corresponds to the fit quality metric for the worst replicate, the line corresponds to 95% confidence interval, with the color corresponding to the fit quality of that replicate. Parameters $k_1'$ and $k_2$ that were identifiable for both replicates are also quite

reproducible, while $k_2'$ and $k_{cyto\_deg}$ are very reproducible and well-defined. The dashed red line indicates the 2-fold reproducibility window. **D** Reproducibility of composite parameter $k_1'/k_2$. Even though $k_1'$ and $k_2$ are well not defined for some genes, their ratio may be highly reproducible and well-defined. The dashed red line indicates the twofold reproducibility window. **E** Comparison of model-inferred half-life with half-life values determined with the actinomycin-D method. The dashed lines link replicates of the model-inferred half-life. The green line indicates a 1:1 relationship and the dashed red line a 2-fold range. We notice a high correlation between the two half-life estimates (Spearman rank correlation of 0.8) but that the model seems to capture a larger range in half-lives than the actinomycin-D method.

(Supplementary Fig. S5B). Some of these RBP motifs were correlated with a higher effective transport rate, e.g., HNRNPK and QKI[13–16]. Conversely, there are some RBP motifs on the 3′-UTR that correlate with a lower effective transport rate, e.g., HNRNPLL[17–19] and YBX1[20–23]. These proteins were shown to be involved in mRNA splicing, transport, and decay.

Next, we examined if the effective transport rate may be affected by the chromatin context of the gene. We measured four histone marks using ChIP-seq in cells prior to stimulation: H3K27ac is associated with active enhancers, H3K4me3 with promoters, H3K36me3 and H3K79me2 with actively transcribed gene bodies[24,25]. The average

peak ChIP-seq signal of all peaks assigned to the closest gene TSS was quantified (see "Methods"), but none of the marks showed a strong correlation with the effective transport rate (Fig. 5D). In addition, machine-learning models were trained to assess if the ChIP-seq signal would add information to the other metrics in predicting the effective transport rate values (see "Methods"). While predicting the effective transport rate using the combination of gene length, intron number, and intron retention significantly improved the prediction over to using just one of these variables, adding ChIP-seq signals in a variety of different windows or measures to those did not increase the predictive power of the models (Fig. 5E).

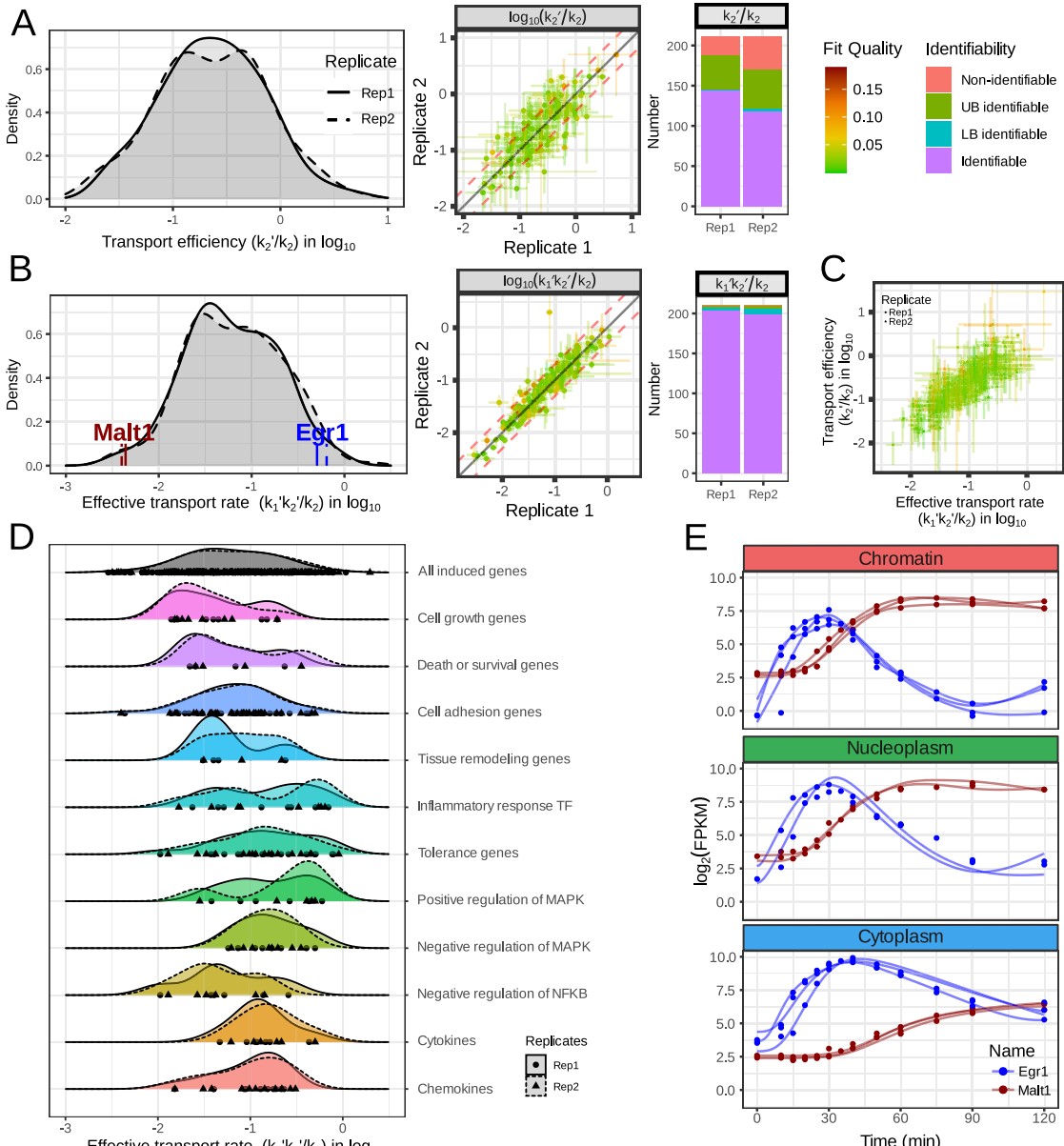

**Fig. 4 | Export rates vary widely across immune response genes. A** Left panel shows the histogram of nucleoplasmic export efficiencies ($k_2'/k_2$) for immune response genes. The distributions are similar for the different replicates and span a 30-fold difference between genes. Middle panel shows the reproducibility of these export efficiency estimates. Right panel shows the number of genes for which this quantity is identifiable. **B** Left panel shows the histogram of effective transport rates ($k_1'k_2'/k_2$) for immune response genes. The distributions are similar for the different replicates and span a 100-fold difference between genes. Middle panel shows the reproducibility of these effective transport rate estimates. Right panel shows the number of genes for which this quantity is identifiable. **C** Correlation of the nucleoplasmic export efficiency and the effective transport rate for the genes for which both parameters are identifiable. **D** The distributions of effective transport rates of genes involved in various biological processes. **E** Example expression for two genes having effective transport rates ($k_1'k_2'/k_2$) on the extremes of the distribution. *Malt1* has a low effective transport rate and *Egr1* has a high rate. From the line graph (top), we noticed that even though *Egr1* is less expressed than *Malt1* in the chromatin fraction, its expression ends up being higher in the cytoplasmic fraction.

## Transport parameters are unaffected by tolerance-inducing pre-stimulation

To further examine whether effective transport rates are unaffected by the genes' chromatin, we produced equivalent experimental datasets in macrophages that had been pre-stimulated with Lipid A and thus rendered into a so-called tolerized state in which the epigenome is substantially altered and gene expression is less responsive to stimulation with the second dose of Lipid A. Similar to the naive condition, replicate data demonstrated high reproducibility (Supplementary Fig. S6A, C), but some genes showed such a low level of expression that they had to be removed from subsequent analyses (Supplementary

Fig. S6B), leaving 186 genes to be fitted. Tolerized macrophages showed slightly increased basal levels for most genes (Fig. 6A, top), but, as expected, they exhibited a strong reduction of induction as observed in the chromatin fraction (Fig. 6A, bottom), and also in subsequent fractions (Fig. 6B).

Likely due to the lower fold gene induction, the fits were not as good as for naive macrophages (see Fig. 6C and Supplementary Fig. 7A, B) with more genes having a fit quality metric of ≥0.06 for both replicates (37 vs. 9 for tolerized vs naive macrophages). However, the number of genes for which parameters were identifiable was similar to the naive condition (Supplementary Fig. S7B top panel), though the

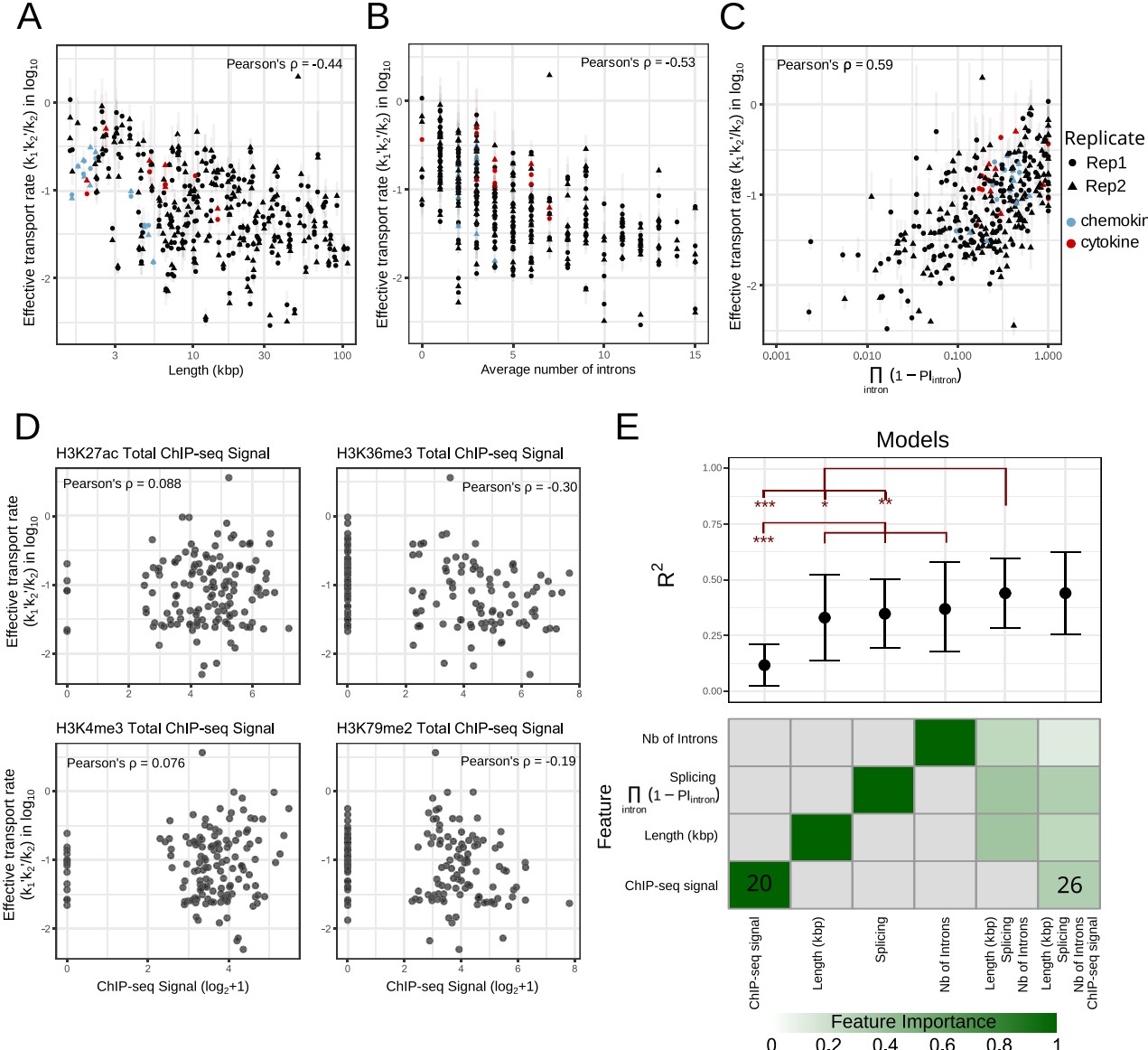

**Fig. 5 | Potential determinants of the effective mRNA transport rate. A** Effective transport rates ($k_1'k_2'/k_2$) show a significant negative correlation with gene length (Pearson's correlation of −0.47 and −0.42 with associated $P$ value of 3e-12 and 9e-10 for replicate 1 and 2, respectively). **B** Effective transport rates show a significant negative correlation with the number of introns (Pearson's correlation of −0.54 and −0.53 with associated $P$ value of 3e-16 and 8e-16 for replicate 1 and 2 respectively). **C** Effective transport rates show a significant positive correlation with splicing probability, averaged over all introns (Pearson's correlation of 0.63 and 0.55 with associated $P$ value <2.2e-16 and of 1e-13 for replicate 1 and 2, respectively). This correlation coefficient is higher than when only the most retained intron is considered (Supplementary Fig. S5A). **D** Sum of ChIP-seq signals of indicated histone

mark associated with the gene do not show a correlation with the effective transport rate (alternatively, windows different sizes along the gene, described in Methods, were tried but yielded no better correlation). **E** Machine-learning models reveal little predictive power in histone modification ChIP-seq signals. Top, plot of $R^2$ values that indicates the predictive power of machine-learning models that consider indicated features. Error bars indicate the mean +/− standard deviation of $R^2$ value of the cross-validation sets. A two-sided $t$ test was used with */**/*** indicating a $P$ value of <0.05, <0.01, <0.001. Bottom, heatmap of the features' importance, defined by the gain in accuracy brought by each feature normalized by the total gain. Numbers in the heatmap correspond to the number of ChIP-seq bins selected by the model.

associated 95% confidence interval was wider for $k_2'$ and $k_{cyto\_deg}$ (Supplementary Fig. S7B middle panel). The estimated parameters for genes with good fits showed good reproducibility in replicates (Supplementary Fig. S7B, bottom panel) and were remarkably similar to the estimated parameters from the naive condition (Fig. 6D); $k_2'$ and $k_{cyto\_deg}$ being mostly within a ±2-fold range. Further, the composite parameters of transport efficiency and effective transport rate for the well fitted genes were close to indistinguishable between naive and tolerized macrophages, with only poorly fitted genes showing some differences (Fig. 6E). This suggests that while the context of chromatin and trans-acting factors regulate transcriptional initiation of immune

response genes, effective transport rates are primarily regulated by context-independent gene structure and sequence features.

**Transport parameters do not regulate the responsiveness but the abundance of mRNA induction**
Two hypotheses address how the mRNA responsiveness of immune response genes is regulated: Intuitively, the effective transport rate, which includes any delays in mRNA processing and splicing, would control responsiveness. The alternative hypothesis posits that cytoplasmic mRNA decay rate determines responsiveness, based on theoretical considerations and actual experimental observations[3,26] in

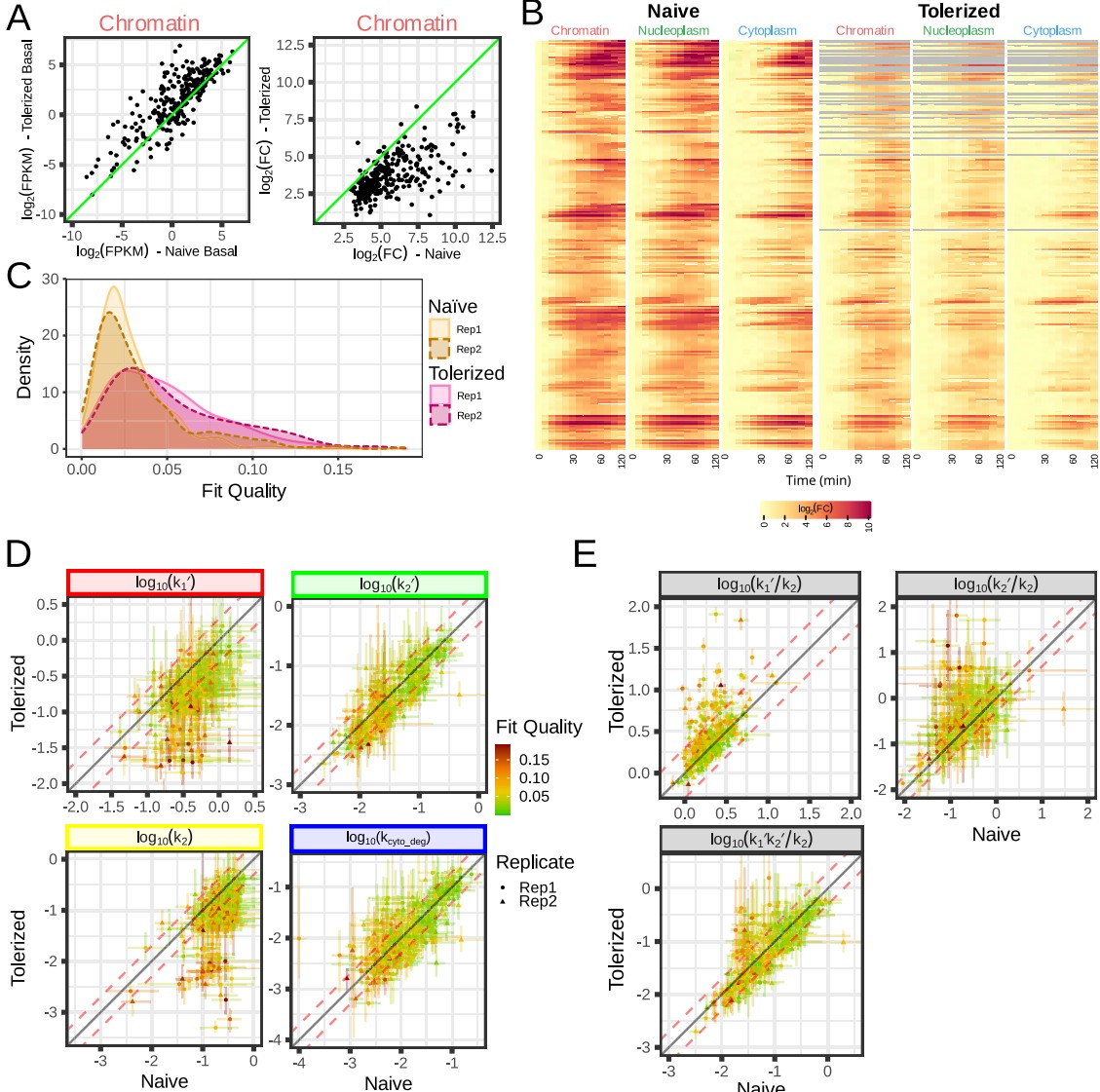

**Fig. 6 | Prior LPS exposure diminishes transcriptional initiation but has little effect on export rates. A** Comparison of observed basal expression (top) and max fold change (bottom) at the chromatin level. Tolerization increases basal expression and decreases induced expression for most genes. **B** Heatmap of fold change in the naive condition and tolerized condition. We observe in all fractions that the induction is dimmed in the tolerized condition. **C** Distribution of fit quality for tolerized and naive macrophages. The fit quality for the tolerized macrophage is not as good as for the naive (heavier right tail). Source data are provided as a Source Data file. **D** Comparison of post-transcriptional parameters in naive and tolerized cells. We observe that $k_1'$ and $k_2$ are not well-defined, as in naive macrophages. However, $k_2'$ and $k_{cyto\_deg}$ are well-defined and very similar between to those in naive condition. **E** Comparison of the composite parameters, effective transport rate ($k_1'k_2'/k_2$), and export efficiency ($k_2'/k_2$), between naive and tolerized conditions. These parameters have similar values for most genes, even if less well-defined in the tolerized condition.

studies that, however, did not consider nuclear-to-cytoplasmic transport. We simulated the gene expression induction of nine hypothetical genes combining a high, medium, or low transport rates, with a short, a medium, and long mRNA half-life (Fig. 7A). The results demonstrated that the mRNA half-life was a primary determinant of responsiveness, which can be quantified as time to half-maximal expression. Examining the control of responsiveness further, we identify regimes in which transport rates may be important. However, plotting the actual transport and degradation rates of immune response genes onto this map, we found that almost all genes fall into the regime where the responsiveness is controlled almost entirely by cytoplasmic mRNA half-life (Fig. 7B). Quantifying the mRNA responsiveness of immune response genes, we observed that it is strongly correlated with their estimated mRNA half-life, but also showed a nonlinear relationship with the effective transport rate (Fig. 7C). Comparing mRNA-degradation rates and effective transport rates we then found an

unexpected correlation (Fig. 7D), with short-lived mRNAs having a higher effective transport rate. We rationalized that the need for rapid responsiveness requires a high cytoplasmic decay rate, which in turn would decrease the magnitude of gene expression; by increasing the effective transport rate, short-lived mRNAs would then be expressed at high cytoplasmic levels (Fig. 7E). Despite the correlation the magnitude of the mRNA-degradation rate tends to be lower than the nuclear export rate, ensuring that it is generally rate limiting. These observations suggest that the effective transport rate is not primarily a determinant of the responsiveness but of the magnitude of gene expression in innate immune responses.

## Discussion

Here, we report that the effective nucleo-cytoplasmic transport rate of immune response genes varies over a 100-fold range. Our measurements were based on triplicate deeply sequenced RNA-seq data from

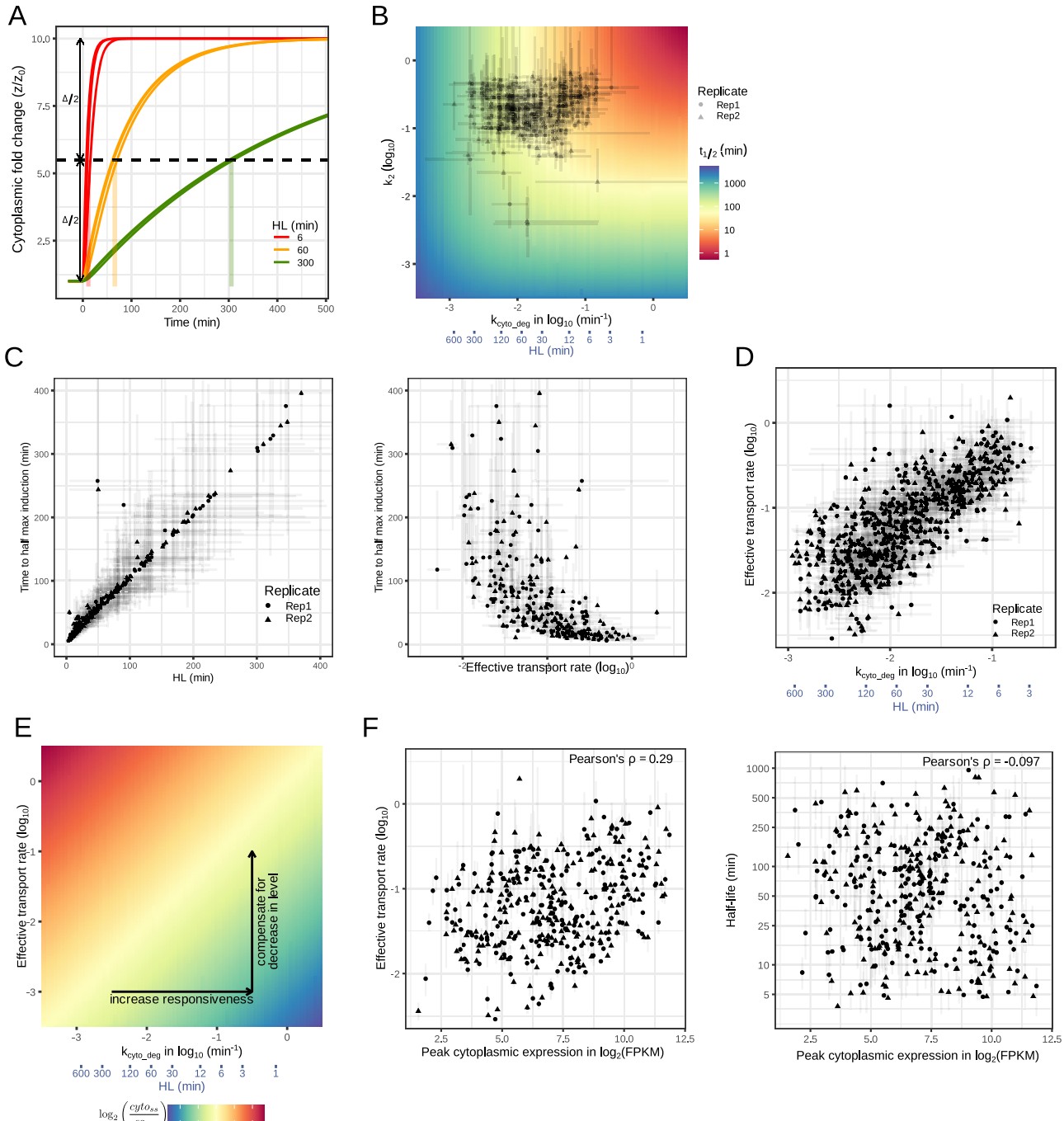

**Fig. 7 | Understanding the link between responsiveness, transport, and mRNA half-life. A** Simulations of gene induction with different parameter values for mRNA decay (with indicated half-lives, HL) and effective nuclear transport rates ranging from 0.03 to 0.3 cytoFPKM/caFPKM min⁻¹ as observed for the fitted genes. Responsiveness is measured as time to half induction. **B** Heatmap of responsiveness for various $k_2'$ and $k_{cyto\_deg}$ parameter values. There are three main regimes, in the top left corner, the responsiveness is determined solely by half-life, in the bottom right corner the responsiveness is solely defined by $k_2$, and the diagonal may depend on both. The parameter valued for the immune response genes are overlaid on top. Most genes fall in the region where responsiveness is solely determined by the cytoplasmic decay rate. **C** Scatterplots to show the correlation of gene responsiveness to a step function with half-life (left panel) and transport (right panel). A strong relationship between half-life and responsiveness is observed. There is a weaker, nonlinear relationship with the transport rate. **D** Scatterplot to show the correlation between the chromatin-to-cytoplasm transport rate and the cytoplasmic decay rates. **E** Heatmap of relative cytoplasmic mRNA abundance as a function the effective transport and cytoplasmic degradation rate. **F** Scatterplots between the peak cytoplasmic expression level of each gene with either the effective transport rate or the half-life of its mRNA. This confirms that neither quantity is correlated with the expression level, supporting the model that the two balance each other to render the level of expression controlled by other mechanisms, such as transcriptional initiation.

chromatin-associated, nucleoplasmic and cytoplasmic compartments, to which a kinetic model of mRNA transport and decay was fit, yielding confidence intervals for the inferred kinetic parameters. Our method leveraged the relative quantitation afforded by RNA-seq without the use of spike-in RNAs, whose addition often introduces variability. Further, estimates for the effective chromatin-to-cytoplasm transport rate had tight confidence intervals, even for genes for which chromatin-release and nucleoplasmic outflux rates (by transport or

decay) were less well definable when mRNA abundances in chromatin and nucleoplasmic fractions were more similar. Thus, the effective transport rates we report could serve for further downstream analysis.

Our regression approach indicated that the effective transport rate may be accounted for to some degree by intrinsic features of the gene such as length and exon/intron structure. In contrast, chromatin context does not seem to be a determinant of effective transport as a range of histone ChIP-seq data showed little predictive power despite testing numerous features, and the altered epigenetic state triggered by prior endotoxin exposure did not alter the kinetic parameters values. While intron retention that results from experimentally perturbing the splicing machinery has been shown to diminish cytoplasmic mRNA[4], our work suggests that it is also contributes to the differences in effective transport seen among immune response genes in wild-type cells. As RBPs mediating export are recruited to exon–exon junction complexes (EJCs), genes that are spliced co-transcriptionally may be favored. But not all introns need to necessarily be co-transcriptionally spliced, as the availability of some EJCs may be sufficient. We do observe that some introns are retained in the nucleoplasm, and that the correlation of transport with splicing is not perfect. Thus, future work may determine whether specific exon–exon junctions are important for recruiting export factors, or whether all exon–exon junctions could in principle make that contribution, or whether TPR pathway may apply, allowing fast transport without EJCs.

An unexpected finding of our analysis was that effective transport rates are highly correlated with cytoplasmic mRNA-degradation rates among immune response genes. This correlation may not be incidental but mechanistically linked: our motif analysis identified a host of RNA-binding proteins that prior literature had implicated in both mechanisms: for example, HNRNPK is involved in splicing[27,28] and shuttles between the nucleus and cytoplasm in a manner consistent with an involvement in mRNA export[29,30] but other work demonstrated its involvement in regulating the mRNA stability of the thymidine phosphorylase gene[31]; further, QKI was shown to be involved in splicing[32] and nuclear export of MBP[33] but other work had implicated it in regulating the mRNA stability of the AIP gene[34].

Why would cytoplasmic mRNA decay be mechanistically linked with effective nuclear mRNA export? Prior work has established that the cytoplasmic mRNA half-life controls the responsiveness of immune response gene expression[3,26]. However, if genes that must be highly responsive to immune signals have evolved a high cytoplasmic mRNA-degradation rate, their expression level would also be dramatically lowered. Mechanistically linking the effective transport rate to cytoplasmic mRNA-degradation allows rapidly induced immune response genes to also be expressed at a high level. Indeed, our modeling analysis shows that the hundred-fold range of effective transport rates may ensure similar levels of expression for both rapidly inducible or long half-life mRNAs. Thus, contrary to expectation, effective transport rates do not directly regulate the stimulus-responsiveness of immune gene expression but regulate the magnitude of gene expression in immune response programs.

## Methods

### Macrophage cell culture and stimulation
Bone-marrow cells were isolated from wild-type C57 BL/6 mice (females, 3 months, approved and maintained by the University of California, Los Angeles Division of Laboratory Animal Medicine accredited by Association for Assessment and Accreditation of Laboratory Animal Care International, AAALAC) and plated with 30% L929-conditioned IMDM 10% serum (Gibco ES) supplemented with Penicillin/Streptomycin, 2-Mercaptoethanol, and L-Glutamine. At day 7, "naive cells" were kept in media for an extra 24 h, while "tolerized cells" were generated by pre-stimulation for 12 h with 100 ng/ml Lipid A, followed by three washes with warm PBS and 12 h rest with

differentiation medium. At day 8, naive and tolerized cells were stimulated with 100 ng/ml Lipid A (Invivogen cat. no. tlrl-mpls) with no change of medium. Three biological replicates were prepared several weeks apart.

### RNA preparation and sequencing
After stimulation, BMDMs were harvested at desired timepoints. Subcellular fractions were prepared as described[8]. Cytoplasmic and nucleoplasmic RNA were isolated using DIRECT-zol micro prep kit (Zymo Research). For chromatin-associated RNA, chloroform extraction was done first and then, the aqueous phase was used to isolate RNA using the DIRECT-zol micro prep kit. In all cases, DNase I digestion was carried out to remove DNA. The nuclear fraction of one replicate was mishandled and thus not available for analysis, however the chromatin and cytoplamic fractions were still used when possible. Strand-specific libraries were generated from 250 ng–1 μg of RNA using KAPA Stranded RNA-seq with RiboErase Library Preparation kit (KAPA Biosystems, Wilmington, MA) according to the manufacturer's instructions. Resulting cDNA libraries were paired-end sequenced multiple times for appropriate depth with a length of 101 bp on an Illumina HiSeq 2000 or Illumina HiSeq4000 (Illumina, San Diego, CA). Replicates run on either platform revealed little technical variability.

### RNA-seq data analysis
Adapter sequences were removed and lower quality 3′-end trimmed if needed using cutadapt v1.12[35]. Reads were aligned to the mm10 genome build with STAR 2.5.2b[36] using Gencode vM14 as reference annotation[37]. Pairs with unmapped reads were filtered out using samtools 1.3.1[38]. Pairs falling onto chrM, chrY and known rRNA regions were also filtered out based on UCSC table browser[39]. Bam files for each sample were merged to a single bam file using samtools[38]. All RNA-seq data (fastqs and bams files) was deposited on ENCODE DCC (https://www.encodeproject.org/awards/U01HG007912/). For gene selection, chromatin-level expression was calculated using featureCounts 1.5.1[40] on the gene whole body for uniquely mapped fragments based on Gencode vM14 annotations[37]. Genes for which all samples had less than 32 fragments were removed from further analysis. Highly expressed genes (3 data points with FPKM ≥1) were kept for further analysis. Then significantly induced genes were selected using R (3.5.1)[41] with edgeR package 3.22.5[42]. Induction had to be at least ten-fold relative to basal and within the first 40 min after LPA stimulation, with FDR corrected $P$ value <=0.01. Non-protein coding genes and predicted genes were filtered out leaving a total of 288 genes for the naive condition. For the tolerized conditions the same list of genes was used if they also had ≥32 fragments in at least one sample and three samples with FPKM ≥1 on the chromatin, leaving 249 out of the 288 for further analysis. Furthermore, sample tracks of chromatin-associated RNA were examined using IGV[43], to filter out false positives due to read-through from a close gene (14 genes for naive and 6 for LPA stimulation, see examples Supplementary Fig. 2). In addition, one gene was removed because the chromatin tracks/junction reads seemed to combine annotated transcripts from a different gene. Thus, we considered 273 genes for the naive condition and 242 for the tolerized condition.

### Actinomycin-D mRNA half-life measurement
Transcription was inhibited by adding 10 μg/ml of Actinomycin D (ActD; A9415, Sigma-Aldrich), at 0, 1, and 3 h of LPA stimulation. Cells were harvested at 0, 30, 60, 90, 120, 240, and 360 min after ActD addition (Supplementary Fig. S4A). Cells were harvested in TRIzol and RNA was isolated using Directzol kit (Zymo) after DNAase treatment. During RNA library preparation, 2 μl of 1:100 diluted RNA spike-in (Ambion ERCC Spike-In Mix Part no 4456740) was added for external normalization. Sequencing was done with a

length of 50 bp on Illumina HiSeq4000 sequencer. RNA sequencing processing included trimming of remaining using cutadapt v1.12[35], alignment of the reads with STAR[36] to mm10 genome with Gencode vM6 annotations[37] supplemented by ERCC spike-in sequences. Unmapped reads were filtered out using samtools[38]. Gene-wise counts were generated with featureCounts[40] using uniquely mapped reads. All sequencing fastq files were deposited to Sequence Read Archive[44] under BioProject IDs PRJNA641336.

We created an R package, called ActDanalyser (https://github.com/dlefaudeux-ucla/ActDAnalyser), which allows users to easily calculate half-life from Actinomycin-D RNA sequencing data. Within the package, functions were implemented to render every step as easy as possible. First, genes with low counts (≤32) in all samples were removed from the analysis. Then the median ratio of each sample's spike-ins to its geometric mean across samples was used as normalization factor and applied to corresponding sample gene set. The library size after normalization decreased with time after actD addition, as expected given that mRNAs were decaying. This helped to flag some experiments that clearly did not follow that pattern and were not included in downstream analysis, for example, the sample corresponding to 120 min after actD addition of the 1 h post Lipid A stimulation experiment (Supplementary Fig. S4B).

To derive mRNA half-life, the following issues were considered: (i) after actD addition, polymerase arrest is not instantaneous. (ii) Late timepoints are less reliable, when short-lived mRNAs are at low levels. Therefore, the regression was implemented to start at any timepoint within the first hour, and the last timepoint was picked such that the regression gives the highest adjusted R² (Supplementary Fig. S4C). The steps are summarized as follows:

- Identify potential start points as: $t \le 1$ h after actD addition and for which the normalized counts (in log$_2$) are not lower than the max of the 1$^{st}$ hour − 0.25× the max decay per hour.
- For each ActD timecourse: (i) identify possible end timepoints, (ii) run linear regression between the log$_2$ normalized counts and time allowing removal of one point as long as at least three timepoints remain. Allow the intercept to be different for each replicate.
- Select the negative slope regression that has the highest adjusted R².
- Calculate slope confidence interval (CI) using the confint function from the R stats package[41]
- Convert slope to mRNA half-life

A web interface, called ActDBrowser (https://www.signalingsystems.ucla.edu/ActDBrowser), was implemented using the R shiny package allowing users to search half-life for specific mRNA(s) in specific cell type and conditions as a resource to the community.

## ChIP-sequencing

ChIP-seq protocol was conducted according to published methods[45] with 5 µg of antibody against H3K4me3 (05-745R, Millipore), H3K36me3 (ab9050, Abcam), H3K27ac (39133, Active-Motif), and H3K79me2 (ab3594, Abcam). ChIP-seq libraries were generated using Kapa Hyper Prep Kit (KAPA Biosystems, Wilmington, MA), and were single-end sequenced on an Illumina HiSeq 2000 (Illumina, San Diego, CA) with a length of 50 bp. FASTQ reads were aligned using the ENCODE-defined analysis pipeline for ChIP-seq read mapping[46,47]. Histone ChIP peaks were called using the ENCODE-defined analysis pipeline for histone ChIP-seq and annotated to the closest gene with HOMER suite v4.11[48]. Biological replicate histone signals were normalized to peak sequence depth using the ENCODE pipeline. Histone mark signals were averaged between replicates. All ChIP-seq data (fastqs files) was deposited on ENCODE DCC (https://www.encodeproject.org/awards/U01HG007912/) and publicly available.

## Regression and machine-learning modeling with ChIP-seq signals

Total, upstream, and downstream histone mark levels were calculated by summing across the mentioned ranges. Alternatively, histone mark levels were partitioned into windows with fixed width based on the average width of each mark. Histone windows were symmetrically centered around the TSS. H3K27ac had four windows with a width of 7500 bp. H3K36me3 had eight windows with a width of 6250 bp. H3K4me3 had four windows with a width of 2500 bp. H3K79me2 had six windows with a width of 8333 bp. For each window, the average was calculated for all ChIP signals located within its bounds. Boundary ranges for windows were manually defined to incorporate a threshold of at least 50% of genome-wide peaks for each histone mark, also taking into account the function of each mark. Thus, for classification, each histone mark had several features associated: fixed width windows, total, upstream, and downstream signal. Extreme Gradient Boosting (XGBtree) models were trained to predict derived mRNA transport parameters using various model with input feature combinations of histone peak features, gene length, number of introns, splicing probability as defined previously with fivefold cross-validation repeated three times and 80/20 data split[49]. Hyperparameters of each model were tuned to improve model R² in the following order−number of rounds, learning rate, maximum depth, child weight, column and row sampling, and gamma. R² metrics were calculated for the resamples of each model and used to compare predictive performance. Plots were generated using the R package ggplot2[50] complexHeatmap[51]. All p-values were determined using the Mann−Whitney $U$ test. Feature importance represent the gain of each feature (calculated with the varimp function from the caret R package) over the total gain of all features (i.e., to sum up to 1). This analysis was done in R.

## Estimating splicing probability

Measurement of nuclear intron percentage used SIRI[12] on the nucleoplasmic RNA-seq data. It measures the number of spliced reads across the junction (EE) and the number of reads spanning the exons-intron junction on both sides of the introns (EI and IE). The percentage of intron (PI) for each individual intron was calculated as:

$$PI = \frac{\frac{EI+IE}{2}}{EE + \frac{EI+IE}{2}} \tag{1}$$

For each gene having a single main isoform, the splicing probability (SP) was calculated assuming that each intron was independently spliced:

$$SP = \prod_{intron} (1 - PI_{intron}) \tag{2}$$

The splicing probability was calculated using the average PIs of the last two timepoints (90 and 120 min) as it correlates well with the basal steady-state PIs but having more reads on the junctions allowing for more accurate quantification. Moreover, the splicing probability was calculated only if the gene had more than ten junction reads (spliced or unspliced) for all its introns.

## RNA-binding protein (RBP) motif analysis

RBP sequence motif analysis used AME tool (Analysis of Motif Enrichment) from the MEME suite[52]. The 5′-UTR and 3′-UTR sequences of each gene main isoform were used and any number of RBP motifs from human and mouse motif databases[53] were searched for. Enrichment was tested by spearman correlation (--method spearman) on the total number of hits (--scoring totalhist) using a threshold of 0.25 times the maximum log odd ratio of the motif to be considered a hit (--hit-to-

fraction 0.25) in the respective UTR sequence versus the effective transport rate derived from the modeling.

## Mathematical model formulation

To describe mRNA transport through the different cellular compartments, a two-step model was written as a system of two ordinary differential equations:

$$\frac{d\text{RNA}_{np}}{dt} = k_{ca \to np} \cdot \text{RNA}_{ca} - \left(k_{np \to cyto} + k_{np\circ}\right) \cdot \text{RNA}_{np} \quad (3)$$

$$\frac{d\text{RNA}_{cyto}}{dt} = k_{np \to cyto} \cdot \text{RNA}_{np} - k_{cyto\_deg} \cdot \text{RNA}_{cyto} \quad (4)$$

This model describes the exact number of RNA transcripts in different cellular compartments. As RNA-seq measurements are only relative, normalization factors $\alpha, \beta, \gamma$ were included:

$$x = \alpha \cdot \text{RNA}_{ca}$$

$$y = \beta \cdot \text{RNA}_{np}$$

$$z = \gamma \cdot \text{RNA}_{cyto}$$

$$\frac{dy}{dt} = \underbrace{\frac{\beta}{\alpha}k_{ca \to np}}_{k_1'} \cdot x - \underbrace{(k_{np \to cyto} + k_{npdeg})}_{k_2} \cdot y \quad (5)$$

$$\frac{dz}{dt} = \underbrace{\frac{\gamma}{\beta}k_{np \to cyto}}_{k_2'} \cdot y - k_{cyto\_deg} \cdot z \quad (6)$$

The value of each normalization factor is the same for all genes, allowing comparisons between genes.

Summary of mathematical model parameters

| Parameter | Description | Unit |
|---|---|---|
| $k_1'$ | Transport rate constant from chromatin to the nucleoplasmic fraction | (npFPKM/ caFPKM)/min |
| $k_2$ | Rate of disappearance from the nucleoplasmic fraction (either by export or degradation) | min$^{-1}$ |
| $k_2'$ | Transport rate constant from nucleoplasmic to cytoplasmic fraction | (cytoFPKM/ npFPKM)/min |
| $k_{cyto\_deg}$ | Cytoplasmic degradation rate | min$^{-1}$ |
| $k_1'/k_2$ | Chromatin-release efficiency | npFPKM/ caFPKM |
| $k_2'/k_2$ | Transport efficiency (from nucleoplasm to cytoplasm) | cytoFPKM/ npFPKM |
| $k_1'k_2'/k_2$ | Effective transport rate (rate constant from chromatin to a cytoplasmic fraction) | (cytoFPKM/ caFPKM)/min |

## Gene annotation for quantifying mRNA abundances

Reliable quantification of mRNA abundances is critical for modeling, and this relies on accurate gene annotations. The model considers full-length transcripts, so to estimate chromatin-associated transcripts we considered the exonic regions within the last 5 kb of each gene. To ensure we have accurate annotation of the transcription end site (TES), genome browser tracks were manually checked and annotated. Gencode annotation and the manual reannotation of the TSS and TES were compared against external databases, using CAGE peaks[54] and 3'-end

sequencing data[10] (Supplemental Fig. S2G). The manual annotation was often closer to the TSS and TES from these external databases than the Gencode annotation[37]. One gene with very low cytoplasmic mRNA levels was removed because it did not allow for reliable identification of the expressed isoform; another gene was removed, for having overlapping transcripts with another gene. Genes for which expression tracks/splice junctions seemed to come from unannotated transcripts were also removed (35 for naive, 33 for Tolerized). Additionally, genes for which the TES location was uncertain were also removed (4 for naive, 3 for Tolerized). Moreover, genes having more than one isoform corresponding to ≥10% and ≥1 FPKM of the total expression in more than ¼ of the samples (custom script in Python 3.7) were considered having multiple isoforms expressed. This curation was based on examining tracks as well as using estimated isoforms expression from cufflinks[55]. When genes were deemed to have multiple isoforms expressed, the last 5 kb was reduced to correspond to the exonic portion that is shared by all expressed isoforms species. If the exonic portion of the last 5 kb region represented less than 500 bp (21 for naive, 19 for Tolerized) genes were removed from further analysis to avoid that the small length undermines the reliability of expression estimation. Overall, 77 genes were removed in naive condition (15 genes based on chromatin RNA filtering and 62 based on cytoplasmic RNA filtering) leaving 211 genes for modeling. In Tolerized conditions, 102 genes were removed (46 based on chromatin RNA filtering and 56 based on cytoplasmic RNA filtering) leaving 186 genes. Details of the manual curation can be found in Supplementary Data 1 and examples in Supplementary Fig. S2. To fit the model, relative gene expression was estimated using FPKM as it is proportional to the number of transcripts.

## Error model for RNA-seq analysis

In order to fit the model to each replicate sample individually, we developed an error model. The main sources of error are measurement error, timepoint sampling error, and biological variability between individual samples. We first considered timepoint sampling error and biological variability between samples. Timepoint sampling error affects highly dynamic gene expression trajectories and therefore is a function of the derivative at that timepoint[56]. Let g be the gene expression and lg be the gene expression in log scale, then $lg_{observed}(timepoint) = lg(t + \Delta t) + \varepsilon_b = lg(t) + \Delta t \cdot lg'(t) + \varepsilon_b$, where $\varepsilon_b$ represents the biological variability and $\Delta t$ the temporal variability. We assume that $\varepsilon_b \sim N(0, \sigma_b^2)$ and $\Delta t \sim N(0, \sigma_t^2)$ and $\Delta t \cdot lg' \sim N(0, slope^2 \cdot \sigma_t^2)$, where $slope = lg'$. Hence $lg_{observed}(timepoint) \sim N(lg(t), slope^2 \cdot \sigma_t^2 + \sigma_b^2)$ and thus $g_{observed}(timepoint) = g(t) \cdot \varepsilon_{total}$, with $\varepsilon_{total} \sim logN(0, \sigma_{total}^2 = \sigma_b^2 + slope^2 \cdot \sigma_t^2)$.

Moreover, mRNA abundance measurements are subject to sampling error, especially when the number of reads for a given gene is small. Sampling error is usually represented by a binomial distribution but given that any given gene will be represented by only a small proportion of the total number of reads (which is large), this can be approximated by a Poisson distribution.

When n is large and the proportion p small: $Binomial(n, p) \simeq Poisson(\lambda = n \cdot p)$, thus $counts_{observed}(timepoint) \sim Poisson(\lambda = N \cdot p)$, where N is the total number of reads and p is the proportion of reads that should belong to the given gene and is proportional to the true mRNA abundance, which is assumed to follow a log-normal distribution. We approximated the log-normal distribution to a Gamma distribution with equivalent mean and variance. Therefore:

$$g(timepoint) \sim logN(lg(t), \sigma_{total}^2)$$
$$\simeq \Gamma\left(k = \frac{1}{\exp(\sigma_{total}^2) - 1}, \theta = (\exp(\sigma_{total}^2) - 1) \cdot \exp\left(lg(t) + \frac{\sigma_{total}^2}{2}\right)\right) \quad (7)$$

This leads to the expression that the observed counts follow a negative binomial:

$$counts_{observed} \sim NB\left(r = \frac{1}{\exp(\sigma^2_{total})-1}, p = 1 - \frac{1}{1+\mu \cdot (\exp(\sigma^2_{total})-1) \cdot \left(\frac{\sigma^2_{total}}{2}\right)}\right)$$
(8)

where μ is the true expected number of counts.

Such a negative binomial distribution is commonly used for representing counts distributions in RNA-seq analyses, for example, in software packages edgeR, DESeq, cuffdiff from cufflinks.

### Cost function for model fitting

The cost function was defined using the likelihood of the model reproducing the $\log_2(FPKM)$ data, with the counts following the negative binomial distribution described above (FKPM are counts normalized for library size and gene length). The total cost sums the negative log-likelihood of each timepoint t of each compartment (cpt):

$$Cost = -\log(\mathcal{L}(\theta,\alpha)\mathcal{D}) \cdot P(\alpha))$$
$$= \sum_t \sum_{cpt} -\log\left(P\left(\left(\log_2\left(FPKM_{cpt}(t)\right)\right)\Big|\theta,\alpha\right)\right) - \log(P(\alpha)) \quad (9)$$

Here, $\theta$ represents the model parameters $k_1'$, $k_2$, $k_2'$, $k_{cyto\_deg}$ and $\alpha$ represents additional parameters, such as for smoothing (spar), time variability ($\sigma_t$) and sample variability ($\sigma_b$) shown below. These were also included in the error model, however, their effects are assumed to be relatively small for a single replicate, thus these parameters were regularized in the cost function by giving them a certain prior that has a higher probability of low values, see below. In one replicate the unstimulated timepoint for the chromatin data was missing; this was added as a parameter ($ca_O$) and also regularized as follows:

Prior distribution used for cost function parameters

| | Prior distribution |
|---|---|
| spar | $\mathcal{N}(\mu=0.45, \sigma^2=0.0025)$ |
| $\sigma_t$ | $\mathcal{N}_{1/2}(\mu=0, \sigma^2=25)$ |
| $\sigma_b$ | $\mathcal{N}_{1/2}(\mu=0, \sigma^2=0.01)$ |
| $ca_O$ | $\mathcal{N}(\mu=ca(t_1)-\Delta ca, \sigma^2=0.25)$ with $\Delta ca = \text{mean}_{r \in rep}(ca_r(t_2)-ca_r(t_1))$ |

where $\mathcal{N}_{1/2}$ represents the half-normal distribution.

### Model simulation

Each replicate was used separately for fitting the model parameters, allowing a comparison of the optimal parameter set. For each replicate, the chromatin-associated expression is interpolated using the.spline function in R (with each point weighted based on its $\log_2(FPKM)$ probability, accounting for sampling error), then the model is simulated using a defined set of parameters. The numerical simulations were done using the deSolve R package[57] as well as the compiler package[41] for faster execution of the ode model and cost function calculation.

### Parameter estimation

Local optimization method (BFGS as implemented in the R optim function) using 1000 different random initialization sets was used to find the best parameter set, as it has been shown to be as efficient as other global methods[58]. The initial parameters were sampled from distributions. This pipeline was done separately for each replicate.

Distributions used to sample the 1000 initial parameter sets

| Parameters | | Distributions |
|---|---|---|
| $\theta$ | $\log_{10}(k_1')$ | $\mathcal{U}(a=-5, b=5)$ |
| | $\log_{10}(k_1'/k_2)$ | $\mathcal{U}(a=-5, b=5)$ |
| | $\log_{10}(k_2/k_2')$ | $\mathcal{U}(a=-5, b=5)$ |
| | $\log_{10}(k_2'/k_{cyto\_deg})$ | $\mathcal{U}(a=-5, b=5)$ |
| $\alpha$ | spar | $\mathcal{N}(\mu=0.45, \sigma^2=0.0025)$ |
| | $\sigma_t$ | $\mathcal{N}_{1/2}(\mu=0.45, \sigma^2=0.0025)$ |
| | $\sigma_b$ | $\mathcal{N}_{1/2}(\mu=0, \sigma^2=0.01)$ |
| | $ca_0$ | $\mathcal{N}(\mu=ca(t_1)-\Delta ca, \sigma^2=0.25)$ with $\Delta ca = \text{mean}_{r \in rep}(ca_r(t_2)-ca_r(t_1))$ |

### Fit quality assessment

To assess the fit quality of parameterized models, we considered that likelihood, used for fitting, is not a sufficient measure. For example, if the fits of two different genes had the same likelihood but for one the simulations were consistently below the data, it would be perceived as worse than the fit for the other gene that is sometimes below and sometimes above the data, especially if the data for the second gene are more jaggedy. Similarly, if one compartment is not well fitted even if the others are, the perceived fit quality would be strongly affected by the data from the poorly fitting compartment. Hence, we developed the following metric which was used to report the perceived fit quality:

$$\max_{cpt}\left(\left|\text{autocorr}\left(error_{cpt}\right)\right| \cdot \frac{\text{mean}\left(|error_{cpt}|\right)}{range_{cpt}}\right) \quad (10)$$

A good fit should have independent residuals, i.e., no autocorrelation and a relatively small remaining error.

To assess the impact of every parameter a profile likelihood approach was used[59]. For each parameter, its profile can be estimated by:

$$PL_{\theta_i}(x) = \max_{\theta|\theta_i=x} \log(\mathcal{L}(\theta|\mathcal{D})) \quad (11)$$

The profile likelihood can also be used to estimate confidence interval of the parameters[59]:

$$CI_{\theta_i,\alpha} = \left\{\theta_i=x| -PL_{\theta_i}(x) \leq \min_\theta(-\log(\mathcal{L}(\theta|\mathcal{D}))) + \frac{1}{2}\Delta(\alpha)\right\} \quad (12)$$

Here, $\alpha$ represents the chosen confidence level. For a sufficient amount of data:

$$\Delta(\alpha) = \text{icdf}(\chi^2_1, \alpha)$$

$$\Delta(0.95) = 3.841459$$

We used the R package dMod[60] as well as the numDeriv package to calculate for each parameter from $\theta$ and combinations such as $k_1'/k_2$, $k_2'/k_2$, and $k_1'k_2'/k_2$ the profile likelihood up to the confidence interval limits (or 1000-fold lower to 1000-fold higher, whichever condition was met first). To be able to apply the profile likelihood measure to the compound parameters and estimate their confidence interval, the model was modified to represent those quantities. Specifically, for $k_1'/k_2$, $k_2'/k_2$ the model was reparametrized as follows, where four

parameters of the model are now $k_2$, $k_1'/k_2$, $k_2'/k_2$, and $k_{cyto\_deg}$:

$$\frac{dy}{dt} = k_2 \cdot \frac{k_1'}{k_2} \cdot x - k_2 \cdot y \tag{13}$$

$$\frac{dz}{dt} = k_2 \cdot \frac{k_2'}{k_2} \cdot y - k_{cyto\_deg} \cdot z \tag{14}$$

With this new parameterization the profile likelihood was employed only for $k_1'/k_2$, $k_2'/k_2$ as the confidence interval was already assessed by the original parametrization for $k_2$ and $k_{cyto\_deg}$

For $k_1'k_2'/k_2$, the model was re-parameterized using the four parameters $k_2$, $k_1'k_2'/k_2$, $k_2'/k_2$, and $k_{cyto\_deg}$:

$$\frac{dy}{dt} = \frac{1}{\frac{k_2'}{k_2}} \cdot \frac{k_1' \cdot k_2'}{k_2} \cdot x - k_2 \cdot y \tag{15}$$

$$\frac{dz}{dt} = k_2 \cdot \left( \frac{k_2'}{k_2} \right) \cdot y - k_{cyto\_deg} \cdot z \tag{16}$$

Similarly, the profile likelihood was employed only for $k_1'k_2'/k_2$ as the confidence interval was already assessed for the other parameters.

### Reporting summary

Further information on research design is available in the Nature Portfolio Reporting Summary linked to this article.

## Data availability

The data that support this study are available from the corresponding author upon reasonable request. All next-gen-sequencing data from fractionated macrophage RNA and from histone modification ChIP are available at ENCODE DCC (https://github.com/ENCODE-DCC, https://www.encodeproject.org/awards/U01HG007912/). ChIP-seq data are available at GEO under Accession numbers: GSE188145, GSE187745, GSE188104, GSE187323, GSE187692, GSE187626. RNA-seq data are available at GEO under Accession numbers: GSE177858, GSE177905, GSE177815, GSE177670, GSE177332, GSE177152, GSE177914, GSE178056, GSE176758, GSE177841, GSE177594, GSE177984, GSE176876, GSE177913, GSE176845, GSE176735, GSE177709, GSE177135, GSE177714, GSE176868, GSE177217, GSE176817, GSE177668, GSE177430, GSE177427, GSE177433, GSE176948, GSE176738, GSE177094, GSE177816, GSE177327, GSE177077, GSE177578, GSE177791, GSE176835, GSE177470, GSE177946, GSE176632, GSE177317, GSE176950, GSE177118, GSE176654, GSE177544, GSE177960, GSE177123, GSE177416, GSE177013, GSE176980, GSE177809, GSE177369, GSE177564, GSE177011, GSE176830, GSE176694, GSE176772, GSE177076, GSE176743, GSE176636, GSE177476, GSE177589, GSE176833, GSE177417, GSE177998, GSE177734, GSE177208, GSE177448, GSE178064, GSE178037, GSE177563, GSE178016, GSE177278, GSE177919, GSE176988, GSE176688, GSE177915, GSE176956, GSE177319, GSE177965. Actinomycin-D next-gen-sequencing data to determine mRNA half-lives are available at https://www.ncbi.nlm.nih.gov/bioproject/PRJNA641336, https://www.ncbi.nlm.nih.gov/Traces/study/?acc=PRJNA641336 with the following accession IDs: SRX8602466, SRX8602467, SRX8602468, SRX8607671, SRX8607672, SRX8607673, SRX8608046, SRX8608047, SRX8608048, SRX8607661, SRX8607662, SRX8610539, SRX8607663, SRX8607665, SRX8607666, SRX8607667, SRX8607668, SRX8607669, SRX8607670, SRX8610540, SRX8610541, SRX8602465, SRX8610048, SRX8610049, SRX8610050, SRX8610518, SRX8610519, SRX8610520, SRX8610521, SRX8610522, SRX8610523, SRX8610524, SRX8610525, SRX8610526, SRX8610527, SRX8610532, SRX8610533, SRX8610534, SRX8610535, SRX8610536, SRX8610537, SRX8610538. Source data are provided with this paper.

## Code availability

Code for the analysis of Actinomycin-RNA-seq data can be accessed at https://github.com/signalingsystemslab/ActDAnalyser. Code for the model-aided analysis of the RNA-seq data from chromatin, nucleoplasm, and cytoplasm can be accessed at https://github.com/signalingsystemslab/mRNA-nuclear-export.

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

## Acknowledgements

This project was a component of the ENCODE Genomics of Gene Reg-ulation Program and supported by U01HG007912 "Ribonomics of Gene Regulation to predict Innate Immune Responses", awarded to A.H., D.B., T.L.J., X.G.X., Y.X., S.S.T., J.E., M.P., and K.S. We acknowledge additional support R01AI127864 (to A.H.). We acknowledge the dedication of all team members and the ENCODE DC team for data deposition.

## Author contributions

Sample and data generation was led and coordinated by S.S. Bioinfor-matic analysis was led by D.L. with contributions by many Ribonomics Team members. Mathematical modeling was undertaken by D.L., chro-matin regression and machine-learning analyses by K.J. A.H. supervised the project. D.L. and A.H. wrote the manuscript with critical contribu-tions from S.S., and Ribonomics Team principal investigators D.B., T.L.J., X.G.X., Y.X., S.S.T., J.E., M.P., and K.S.

## Competing interests

The authors declare no competing interests.

## Additional information

## the UCLA Ribonomics Group

Alexander Hoffmann [1,2,3] ✉, Diane Lefaudeux[1,2], Supriya Sen[1,3], Kevin Jiang[1,2], Jose Guillermo Sanchez Arriola[2], Nick Miller[2], Zhang Cheng [1,2], Emily Yi Hsin Chen[1], Sukanya Roy[1], Roberto Spreafico [1,2], Tracy L. Johnson[3,4], Erin M. Wissink [3,4], Shubhamoy Ghosh[3,4], Douglas L. Black[1,3], Chia-Ho Lin[1,3], Xinshu Xiao[2,3,5], Jae Hoon Bahn[2,5], Ashley A. Cass[2,5], Esther Y. H. Hsiao[2,5], Stephen T. Smale[1,3], Jerry Hung-Hao Lo[1,3], Jason Ernst [2,3,6], Artur Jaroszewicz[2,6], Matteo Pellegrini[2,3,4], Marco Morselli[2,3,4], Yi Xing[1,2,3], Eddie Park[1,2] & Sri Kosuri [2,3,7]

[4]Department of Molecular, Cell, and Developmental Biology, University of California, Los Angeles, Los Angeles, CA 90025, USA. [5]Department of Integrative Biology and Physiology, University of California, Los Angeles, Los Angeles, CA 90025, USA. [6]Department of Biological Chemistry, University of California, Los Angeles, Los Angeles, CA 90025, USA. [7]Department of Chemistry and Biochemistry, University of California, Los Angeles, Los Angeles, CA 90025, USA.

