## [Peer Review File · Nature Communications]

REVIEWER COMMENTS

Reviewer #1 (Remarks to the Author):

This is an interesting manuscript that attempts to address the question of nuclear export rates of innate immune response genes following an external stimulus (in this case, Lipid A). This is a smart approach, given that the dataset is tractable and the number of analysed genes is large enough to enable somewhat general conclusions.

I do however have a number of concerns regarding the interpretation and conclusions of the study that should be resolved before publication.

1.. How general are their findings? I realise that the authors have been careful in not extrapolating their claims on export of immune response genes to other subtypes of RNAs, but are their conclusions applicable to other RNAs - e.g. highly expressed transcripts, or more lowly expressed transcripts such as those with a lower expression in the cytoplasm? How does the steady state expression level of the transcript in the cytoplasm dictate the export efficiency? In other words, for transcripts with a high relative level of expression in the cytoplasm (e.g. Cd74) in unstimulated samples, does this affect the transport rate?

2. I am confused with the interpretation of Figure 6 - how can the authors say that prior LPS exposure diminishes transcriptional initiation but has little effect on export rates? There appear to be significant differences in effective transport rate ($k_1'k_2'/k_2$) and export efficiency (k_2'/k_2) between naive and tolerized conditions - especially given the lines are not linear. This is in fact to be expected, I would be surprised if this wasn't the case. The authors should have a look at newly synthesised transcripts with 4sU labelling to circumvent this issue. Do they show different kinetics of export?

Minor comments:

1. Why do the authors find that short lived mRNAs having a higher effective transport rate is unexpected? This is an interesting finding that makes biological sense.

2. In general, while I appreciate the authors willingness to explain their methodology, it is rather dense to read and difficult for people who are not computational biologists to understand - this is especially true for a general interest journal such as Nature Communications.

Reviewer #2 (Remarks to the Author):

Lefaudeux and colleagues use stimulation of immune genes in mouse bone marrow derived macrophages as a model to study how nuclear-to-cytoplasmic transport contributes to stimulus responsiveness. They generate rich, tightly-timed RNA-seq datasets from chromatin, nucleoplasmic and cytoplasmic fractions and use them to fit and validate the parameters of differential equations describing the rates of transport and degradation. Among the roughly 200 induced genes, they observe up to ~100-fold differences in transport rates. While they find that responsiveness is mostly dependent on cytoplasmic decay rates rather than transport, they do determine that genes with high rates of decay have highly effective transport, possibly to facilitate equal expression levels. The authors further observe that transport rates correlate with gene length and number of introns but not with chromatin features (but see below).

This work provides solid insight into the role of transport for induction of immune cells and based on an outstanding dataset and judicious curation. The authors are requested to address several minor points:

1. Why are only two of the three replicates shown in most figures? If there was a reason to exclude a

- particular replicate, what was the rationale and why was a filtering criterion of a minimum of 32 reads per gene for all three replicates required?
2. How were overall parameters for jointly for all replicates obtained?
 3. How were compound parameters obtained in cases where the single parameters could not be fitted?
 4. For the interpretation of the functional implications of correlates of post-transcriptional for transport, does the number of introns or the length of the gene drive the correlation?
 5. To compare the degree of nucleoplasmic splicing with transport rate (p. 13), the authors make the assumption that splicing of multiple introns within the same transcript is independent. However, it has been shown (e.g., Drexler et al., Mol Cell 2020) that splicing is coordinated between adjacent introns. Does retention of the most retained intron correlate similarly with effective transport rate?
 6. It is not clear that the methodological choices made to calculate ChIP signal for each gene, i.e. assigning peaks to the nearest TSS and averaging peak signals, result in a meaningful scoring of a gene's chromatin environment or conversely may be the reason why no correlation was found. It may be argued, for example, that the chromatin environment near the PAS may correlate with release from chromatin. The authors should attempt to compare transport rates with the general chromatin environment around the gene as well as at the PAS before making the claim that no correlation exists. For the existing method, the window sizes for averaging ChIP signals around TSS should be provided and the method of averaging better explained in the Methods.
 7. The authors characterize the parameters fitted in tolerized conditions as very similar to those from naïve conditions. However, there appear to be groups of genes (albeit with relatively lower fit quality) that have lower k_1' and k_2 , as well as higher k_1'/k_2 and k_2'/k_2 in tolerized conditions. How do the authors interpret these differences?
 8. Can levels of intron retention serve as a proxy for decay rate via its correlation with the transport rate?
 9. Fig. 1D: Add numbers of expressed genes and highly expressed genes after first filtering steps for a more comprehensive picture of how representative the filtered genes are.
 10. Fig. 3CD: Explain units/legend and red dashed lines in D.
 11. Main text, p. 9: Reference to Supplementary Figure 3 should be 4.
 12. Main text, p.15: Reference to Supplementary Figure S7C should be B.
 13. Suppl. Fig. S3C: legend missing
 14. Fig. 7D-E: x-axis labels missing

Reviewer #3 (Remarks to the Author):

Summary:

This study characterized mRNA nuclear export efficiency in endotoxin-stimulated murine macrophages where export rates varied almost 100-fold and correlates with gene context/sequence features. The authors isolated RNA from three subcellular locations (chromatin-associated, nucleoplasmic, and cytoplasmic) at twelve timepoints spanning a total of 2 hours after stimulation by Lipid A, and filtering for strongly inducible genes led to 273 genes. Model fitting led to the estimation of rates of transport between cellular subcompartments and cytoplasmic degradation rates of these genes. Extreme Gradient Boosting (XGBtree) models were trained on this data to predict these mRNA transport parameters and achieved a mean R^2 of a little under 0.5 using gene length, splicing probability in the nucleoplasm, and number of introns. Generally suggesting that longer genes with more introns and less efficient splicing in the nucleoplasm are less efficiently transported to the cytoplasm. The authors found correlation between these initial experiments and actinomycin D experiments which inhibits transcription. The authors repeated the Lipid A stimulation time course experiments on tolerized macrophages and found these results on 186 genes to be similar to the naïve condition's composite parameters of transport efficiency, suggesting that context-independent gene structure and sequence primarily influence effective transport rates between the subcompartments.

Major points:

1. "Error model and cost function" in Methods:

- a. I'm not following the approximation of total error. How does the slope term arise?
- b. I'm also not following the steps leading to the negative binomial definition of counts.
- c. Could we ask for a more detailed step-by-step derivation of these two definitions in their response to reviewers?

2. Fig S1D: Why does the nucleoplasmic correlation heatmap have white "stripes" in the plot? Based on the size of the black lines along the axes, it also looks like some time points are not reported with all the triplicate data? Ex. Singleton to the right of the label "30" on the x-axis.

3. Related to Fig 2: Why was the modeling done with 2 replicates if 3 replicates were collected?

4. Fig 2, 3, 6, S7: k_{deg} in these figures should be notated as $k_{(cyto\ deg)}$ to keep consistent with Figure 1 and the Methods section and not confuse the reader into thinking it is the nuclear and/or cytoplasmic degradation rate(s). Please also change related Main Text referencing k_{deg} .

5. Fig 3E: How were model-inferred half-lives calculated? From $k_{(cyto\ deg)}$ alone or $k_{(cyto\ deg)}$ and $k_{(np\ deg)}$?

6. Could the authors provide a specific section in "Methods" for Figure 7's simulations? It is hard to evaluate the authors' conclusion that responsiveness is largely due to mRNA half-life, and if the mRNA half-life parameter they are referring to is derived from only cytoplasmic decay or both nuclear and cytoplasmic decay. Also, what if any role does transcription rates play in these simulations?

Minor points:

1. Some figures look to have lower resolution compared to others. This may be from the upload process? Ex. Fig 1, Fig 2 vs. Fig 3

2. Please use the prime character ' instead of the apostrophe character ` in 3' and 5'.

3. "Error model and cost function" in Methods: the acronym "cpt" is not defined and assumed to mean "compartment".

4. Fig S4C: What is the difference between unfilled circles and X's in this plot?

5. Fig S4D: "Dashed lines represent the 95% confidence interval..." means the gray dashed lines? The next sentence refers to the red dashed lines.

6. Fig S5: The "A" panel notation is cut off. Should it be removed since there are no other panel elements?

7. Fig S7B: Typo of "Confidence Interval" in subplot title.

Reviewer #4 (Remarks to the Author):

Lefaudeux et al. propose a mechanistic model of mRNA transport from chromatin to nucleoplasm to cytoplasm for immune response genes in mouse. The analysis is based on very interesting RNA-seq data measuring mRNA quantity at quite high temporal resolution in the three compartments. The fitted kinetic model enables the estimation of several rates related to the appearance, disappearance and transport of mRNA from one compartment to another, for each gene individually. Results show that transport rates from chromatin to cytoplasm greatly vary from one gene to another, and seem to be essentially linked to gene length and the number of introns, but not to the chromatin states (histone marks) of the genes.

The question addressed in this paper and the data produced are very interesting IMO. The kinetic model is sound and the methodology developed to assess the model and conclusions make sense. However, as a statistical machine learning researcher, I didn't succeed to completely understand the mathematics developed to estimate the parameters of the differential equations. From what I understand, the methodology used in the paper may be quite usual for the physicist community, but it is quite different from what is done in the ML literature. As a consequence, I don't know if this is really helpful, but here are some points I have raised:

- in the ML literature, one of the first assessment (and certainly the most important) that is done when a model has been trained is to estimate the error of the model on an independent test set, i.e. on a set of examples that have not been used to estimate the parameters of the model. Very surprisingly I didn't see this important verification here. The authors provide some comparison between real data and simulated data, and even provide a measure of fit quality, but from what I understand these comparisons were done directly with the data used to estimate the parameters of the model. I suppose that such a test on independent data could also be done here, for example on some time points left out before training (maybe using a cross-validation procedure), or using the data of another replicate.
- concerning the fit quality statistics, I don't understand the sentence "As the negative log likelihood is not easily compared between genes given that it also depends on the expression level". Please explain what do you mean, and why you cannot use directly the cost function used for training as a fit quality index.
- In part Method, the section "Error model and cost function" is not clear. I don't understand the logic between the different equations, and the final expression of the likelihood is cryptic for me. As the final goal is a regression problem, I don't understand why the cost function does not involve a term related to the classical mean square error. Also, please number the formulae and refer to these numbers in the text.
- At several places in the draft the authors mention "the likelihood of the data". This doesn't make sense for me: the data is given, so it cannot be "likely". The likelihood refers to the model, not to the data.
- In the RBP motif analysis, the reported p-values are quite high (not very good). Did they have been corrected for multiple testing problem? Also, please provide additional information: which motif database has been used exactly? Which were the parameters of AME? etc.
- For some figures the resolution is very bad and the text inside these figures is almost unreadable.

Reviewer #5 (Remarks to the Author):

The paper explore mRNA transport and degradation for genes associated to the innate immune response. The overall analysis consider 3 subsequent states of each mRNA, from chromatin, to nuclear envelope to cytoplasmic mRNA for finally to be degraded. The analysis is thorough and contains usefull quantitative lessons on typical rates of mRNA in mammalian cells..

I recommend publication, with some clarifications:

- 1) Rate parameters are presumably in 1/minutes? (couldnt find that in paper)
- 2) Fig. 3 D. maybe explain why k_1'/k_2 is interesting? (i.e. indicate how much there is on npRNA to cRNA)
- 3) Fig. 4A and B please use same x axis for easier comparison. Maybe even put panel E below these panels with same x-axis to allow easier comparisons.
- 4) Fig 7B) indicate that k_{deg} is about a factor 1 to 10 slower than k_2 , indicating that degradation typically is slower than last step in transport. However it could still be faster than whole transport(effective transport rate).
- 5) The whole discussion associated to fig 7 seems equivalent to the old fact that if x is produced according to $dx/dt=a-x*k_{deg}$ then k_{deg} sets the time to reach half max while A/k_{deg} sets the max level. Thus it is not surprising that k_{deg} sets responsiveness (Fig. 7C trivial)
- 6) In Fig. 7D,E,F) what is relative transport? Another name for effective transport? Panel D is interesting, but please explain better, perhaps using $dx/dt=a-x*k_{deg}$ with A =transport roughly scaling with k_{deg} .

Overall, what about total level of mRNA in first compartment (RNAc)? Eg. does total mRNA level correlate with any of the quantities in Fig. 7D? Or does total level of final produced protein in cytoplasm?

Kim Sneppen

REVIEWER COMMENTS

Reviewer #1 (Remarks to the Author):

This is an interesting manuscript that attempts to address the question of nuclear export rates of innate immune response genes following an external stimulus (in this case, Lipid A). This is a smart approach, given that the dataset is tractable and the number of analysed genes is large enough to enable somewhat general conclusions.

We are pleased with the reviewer's interest in the topic and our approach, and their appreciation of quality of the work.

I do however have a number of concerns regarding the interpretation and conclusions of the study that should be resolved before publication.

1.. How general are their findings? I realise that the authors have been careful in not extrapolating their claims on export of immune response genes to other subtypes of RNAs, but are their conclusions applicable to other RNAs - e.g. highly expressed transcripts, or more lowly expressed transcripts such as those with a lower expression in the cytoplasm? How does the steady state expression level of the transcript in the cytoplasm dictate the export efficiency? In other words, for transcripts with a high relative level of expression in the cytoplasm (e.g. Cd74) in unstimulated samples, does this affect the transport rate?

These are interesting questions. As the reviewer remarked, our approach leverages the inducibility of gene expression as a way to measure kinetics. Therefore we cannot make statement about the export efficiency of constitutively expressed genes. As we show in Figure 7 that export efficiency and cytoplasmic mRNA half-life are anti-correlated, we examined the mRNA halflife distribution of constitutively expressed genes. We found that is almost as wide as for induced genes (Fig. S4E). That would suggest that the export efficiency is also widely distributed for constitutively expressed genes. Interestingly, for induced genes, the level of expression is not well correlated with mRNA halflife or export efficiency. This is now shown explicitly in a new Figure 7F. Their anticorrelation ensures that overall neither metric is necessarily a determinant of expression level, though of course, for an individual gene one or the other may be a key determinant.

2. I am confused with the interpretation of Figure 6 - how can the authors say that prior LPS exposure diminishes transcriptional initiation but has little effect on export rates? There appear to be significant differences in effective transport rate ($k_1'k_2'/k_2$) and export efficiency (k_2'/k_2) between naive and tolerized conditions - especially given the lines are not linear. This is in fact to be expected, I would be surprised if this wasn't the case. The authors should have a look at newly synthesised transcripts with 4sU labelling to circumvent this issue. Do they show different kinetics of export?

The $\log_{10}(k_1'k_2'/k_2)$ values of naïve and tolerized conditions are within 2-fold (red dashed lines). This is similar to the variability between replicate samples of the naïve condition (Figure 4B). The $\log_{10}(k_2'/k_2)$ values of naïve and tolerized differ slightly more, but the variability between naive replicates was also higher (Figure 4A). A few genes stand out as having very

different values between naïve and tolerized condition; however, for those genes the model does not fit the data well as denoted by the more orange/red color. We have mentioned this in the text.

We'd also like to clarify that these values are rate constants that are independent of throughput or flux. In our study we have used an optimized protocol to measure chromatin-associated RNA, as pilot studies showed that it is more reliable than 4sU labeling method. caRNA measurements indeed show a major reduction in nascent mRNA synthesis in the LPA-tolerized conditions. Thus, in tolerized conditions there is less transport flux because there is less synthesis of nascent transcripts, but the rate constants are independent of flux.

Minor comments:

1. Why do the authors find that short lived mRNAs having a higher effective transport rate is unexpected? This is an interesting finding that makes biological sense.

We are glad that the reviewer feels that the correlation between high effective transport and fast cytoplasmic degradation makes biological sense. We also feel it makes a lot of sense, as we show in Figure 7.

However, when we started the study we did not expect to find that the transport efficiency varies so widely, over a hundred-fold range, such that for some genes <1% of newly made nascent mRNA actually makes it to the cytoplasm. What was unexpected is that the transport efficiency varies so much, given that there is so little indication in the literature for this.

What is also unexpected that the transport rate constants do not contribute to the response time of the inducible gene, as prior papers suggested this (Pandya-Jones et al 2013, PMID: 23616639).

We show that the response time is entirely determined by its mRNA half-life.

2. In general, while I appreciate the authors willingness to explain their methodology, it is rather dense to read and difficult for people who are not computational biologists to understand - this is especially true for a general interest journal such as Nature Communications.

We have tried to reduce the detail in the next and put it into the Materials & Methods instead.

Other reviewers asked for further detail. We hoped to provide the right balance.

Reviewer #2 (Remarks to the Author):

Lefaudeux and colleagues use stimulation of immune genes in mouse bone marrow derived macrophages as a model to study how nuclear-to-cytoplasmic transport contributes to stimulus responsiveness. They generate rich, tightly-timed RNA-seq datasets from chromatin, nucleoplasmic and cytoplasmic fractions and use them to fit and validate the parameters of differential equations describing the rates of transport and degradation. Among the roughly 200 induced genes, they observe up to ~100-fold differences in transport rates. While they find that responsiveness is mostly dependent on cytoplasmic decay rates rather than transport, they do determine that genes with high rates of decay have highly effective transport, possibly to facilitate equal expression levels. The authors further observe that transport rates correlate with gene length and number of introns but not with chromatin features (but see below).

This work provides solid insight into the role of transport for induction of immune cells and based on an outstanding dataset and judicious curation.

We're pleased with the reviewer's appreciation for the quality of the dataset and the question being addressed with it.

The authors are requested to address several minor points:

1. Why are only two of the three replicates shown in most figures? If there was a reason to exclude a particular replicate, what was the rationale and why was a filtering criterion of a minimum of 32 reads per gene for all three replicates required?

The nucleoplasmic fraction samples of replicate 3 were mishandled and hence did not produce reliable data. However, we still felt that the chromatin and cytoplasmic fraction samples were valuable as they could be used for some aspects of the analysis: i.e. for the gene selection in Figure 1, and when the parameters jointly to the 3 replicates in Figure 2. We have clarified this in the Methods.

2. How were overall parameters for jointly for all replicates obtained?

A joint fitting was done to find a single parameter set that would fit all replicates (Figure 2). For that joint fitting, each replicate had its own chromatin data as input but the model would be run with the same parameter set for all replicate data and the joint cost function would sum the individual replicate cost for their nucleoplasmic (available for two replicates) and cytoplasmic data (available for three replicates). Thus, a common parameter set would be derived.

The model was also used and fitted on each of the two complete replicates separately to derive a parameter set for each replicate (Figure 3, 4). Separate parameter sets provided information about the variability and confidence in the derived parameter values.

We have clarified this process in the text and the Methods.

3. How were compound parameters obtained in cases where the single parameters could not be fitted?

The initial model is a straight-forward description of the biochemical processes of synthesis, decay, and transport. However, we found that not all parameters could be derived from the available measurements. We then algebraically reparametrized the equations using compound parameters in the hope that the compound parameter had a narrower confidence interval. (If a ratio of 2 variables is fixed, it is not possible to specify the value of each variable, unless more information is provided.) How the equations were manipulated is described in the Methods.

4. For the interpretation of the functional implications of correlates of post-transcriptional for transport, does the number of introns or the length of the gene drive the correlation?

We apologize for the brevity of our description. In fact, different panels in Figure 5 describe this. 5A shows that the effective transport rate shows significant negative correlation with gene length. 5B shows that it also shows significant negative correlation with number of introns. 5C shows that the absence of intronic reads in nucleoplasmic RNA is correlated with fast effective transport. However, we can't infer causality from correlation, thus we refrain from identifying the causal driver.

5. To compare the degree of nucleoplasmic splicing with transport rate (p. 13), the authors make the assumption that splicing of multiple introns within the same transcript is independent. However, it has been shown (e.g., Drexler et al., Mol Cell 2020) that splicing is coordinated

between adjacent introns. Does retention of the most retained intron correlate similarly with effective transport rate?

We thank the reviewer for this question. Indeed, we have undertaken this analysis also: When comparing the effective transport rate to the intronic reads of the most retained intron we also found a correlation. However, the correlation was less strong, maybe because considering a single event (and selecting for the highest number) renders the analysis more sensitive to measurement error. This analysis is shown in Supplementary Figure S5A.

6. It is not clear that the methodological choices made to calculate ChIP signal for each gene, i.e. assigning peaks to the nearest TSS and averaging peak signals, result in a meaningful scoring of a gene's chromatin environment or conversely may be the reason why no correlation was found. It may be argued, for example, that the chromatin environment near the PAS may correlate with release from chromatin. The authors should attempt to compare transport rates with the general chromatin environment around the gene as well as at the PAS before making the claim that no correlation exists. For the existing method, the window sizes for averaging ChIP signals around TSS should be provided and the method of averaging better explained in the Methods.

We apologize for not describing the analysis correctly. Contrary to our original description, Figure 4D actually shows an analysis of the sum of all reads associated with the gene (from TSS to PAS) from the indicated histone mark ChIP-seq. We have corrected this. We now also mention alternate analyses (that also did not yield better results) using window sizes of 1kb or 5kb along each gene. We would like to point out that the machine learning model analysis involves several different measures, including various windows, to identify the most predictive combination. That is also described more clearly now.

7. The authors characterize the parameters fitted in tolerized conditions as very similar to those from naïve conditions. However, there appear to be groups of genes (albeit with relatively lower fit quality) that have lower k_1' and k_2 , as well as higher k_1'/k_2 and k_2'/k_2 in tolerized conditions. How do the authors interpret these differences?

The reviewer is right to point out a few genes that show differences in these values.

But as the reviewer said these genes have lower fit quality. We therefore did not want to overinterpret results that could be artifact from those lower quality fits. We have added a clause about these poorly fitted genes within the text.

8. Can levels of intron retention serve as a proxy for decay rate via its correlation with the transport rate?

We agree with the reviewer that that is a likely interpretation given that a poor effective transport rate is correlated with the presence of intronic reads in nucleoplasmic mRNA. However, the correlation is not super strong and there are other factors. Hence, intron retention is an imperfect proxy for nucleoplasmic decay rate.

9. Fig. 1D: Add numbers of expressed genes and highly expressed genes after first filtering steps for a more comprehensive picture of how representative the filtered genes are.

We thank the reviewer for this suggestion. Gene numbers are now included in Fig. 1D.

We thank the reviewer for pointing out the following errors and we have fixed them.

10. Fig. 3CD: Explain units/legend and red dashed lines in D.
11. Main text, p. 9: Reference to Supplementary Figure 3 should be 4.
12. Main text, p.15: Reference to Supplementary Figure S7C should be B.
13. Suppl. Fig. S3C: legend missing
14. Fig. 7D-E: x-axis labels missing

Reviewer #3 (Remarks to the Author):

Summary:

This study characterized mRNA nuclear export efficiency in endotoxin-stimulated murine macrophages where export rates varied almost 100-fold and correlates with gene context/sequence features. The authors isolated RNA from three subcellular locations (chromatin-associated, nucleoplasmic, and cytoplasmic) at twelve timepoints spanning a total of 2 hours after stimulation by Lipid A, and filtering for strongly inducible genes led to 273 genes. Model fitting led to the estimation of rates of transport between cellular subcompartments and cytoplasmic degradation rates of these genes. Extreme Gradient Boosting (XGBtree) models were trained on this data to predict these mRNA transport parameters and achieved a mean R^2 of a little under 0.5 using gene length, splicing probability in the nucleoplasm, and number of introns. Generally suggesting that longer genes with more introns and less efficient splicing in the nucleoplasm are less efficiently transported to the cytoplasm. The authors found correlation between these initial experiments and actinomycin D experiments which inhibits transcription. The authors repeated the Lipid A stimulation time course experiments on tolerized macrophages and found these results on 186 genes to be similar to the naïve condition's composite parameters of transport efficiency, suggesting that context-independent gene structure and sequence primarily influence effective transport rates between the subcompartments.

We thank the reviewer for this concise summary.

Major points:

1. "Error model and cost function" in Methods:

a. I'm not following the approximation of total error. How does the slope term arise?

To be able to fit a single replicate we need to model the error associated with the experimental measurement. We take into account two main types of error, the first being measurement error, i.e. error between the measured value and the true value; the second being time point error, i.e. error between the time when the sample should have been taken and the actual time it was taken. For this second type of error the slope is important. Intuitively, if what we measure does not change with time it does not matter if the experimental time point is actually measured few minutes later than it is supposed to be. However, if what we measure is highly dynamic (high positive or negative slope), the measurement is highly affected by timepoint errors. We have substantially revised and improved the description of the error model.

b. I'm also not following the steps leading to the negative binomial definition of counts.

We have substantially revised and improved the description of the part of the methods.

c. Could we ask for a more detailed step-by-step derivation of these two definitions in their response to reviewers?

We apologize for the lack of clarity. These explanations of this part of the methods has been improved and more details in the derivation has been added.

2. Fig S1D: Why does the nucleoplasmic correlation heatmap have white “stripes” in the plot? Based on the size of the black lines along the axes, it also looks like some time points are not reported with all the triplicate data? Ex. Singleton to the right of the label “30” on the x-axis. In total, 3 replicate experiments were undertaken, but in one replicate the nucleoplasmic fraction samples were mishandled and no data is available for them: hence the white stripes in Fig. S1D. This third replicate was still used when possible (for example for selection of induced genes on the chromatin fraction, and for joint fitting of the model to all data). Here it is shown for transparency. Most time points have 2 or 3 replicates, but a few time points have only one replicate like the 35 min as is mentioned. Given that our results are from applying the model to all timepoints, not one timepoint at a time, such inconsistencies in timepoint availability do not affect the conclusions.

3. Related to Fig 2: Why was the modeling done with 2 replicates if 3 replicates were collected? As described to Reviewer 2, the third replicate lacked data from the nucleoplasmic fraction, which rendered some parameters unidentifiable (even theoretically) based only on chromatin-associated and cytoplasmic mRNA data. However, datasets could be used for the joint fitting in conjunction with the other replicates. We have clarified this in the methods.

4. Fig 2, 3, 6, S7: k_{deg} in these figures should be notated as $k_{(cyto\ deg)}$ to keep consistent with Figure 1 and the Methods section and not confuse the reader into thinking it is the nuclear and/or cytoplasmic degradation rate(s). Please also change related Main Text referencing k_{deg} . We thank the reviewer for this suggestion and took it into consideration by modifying all corresponding figures.

5. Fig 3E: How were model-inferred half-lives calculated? From $k_{(cyto\ deg)}$ alone or $k_{(cyto\ deg)}$ and $k_{(np\ deg)}$? The model-inferred half-lives are for the cytoplasmic mRNA, i.e. $k_{(cyto\ deg)}$. We have used this term throughout the revised manuscript.

6. Could the authors provide a specific section in “Methods” for Figure 7’s simulations? It is hard to evaluate the authors’ conclusion that responsiveness is largely due to mRNA half-life, and if the mRNA half-life parameter they are referring to is derived from only cytoplasmic decay or both nuclear and cytoplasmic decay. Also, what if any role does transcription rates play in these simulations?

We thank the reviewer for the question. We have now clarified that we are exclusively referring to cytoplasmic mRNA half-life. In Fig. 7, we show that the cytoplasmic decay is the primary determinant of the response time, and that the effective transport rate only makes a minor contribution. Transcription rate does not determine responsiveness as responsiveness is defined

by time to half maximal. Transcription rates determine the maximum level, but not the time to reach that level.

Minor points:

1. Some figures look to have lower resolution compared to others. This may be from the upload process? Ex. Fig 1, Fig 2 vs. Fig 3

We apologize for this, which seems to have been the result of embedding the figures in a single document. In the revised submission we kept them separate.

2. Please use the prime character ' instead of the apostrophe character ‘ in 3' and 5'.

We thank the reviewer for pointing this out; we have corrected it.

3. “Error model and cost function” in Methods: the acronym “cpt” is not defined and assumed to mean “compartment”.

Yes, it is meant for compartment. Thank you for pointing that out we added the acronym to the relevant method section.

4. Fig S4C: What is the difference between unfilled circles and X's in this plot?

Thank you for pointing that out, the corresponding legend has been adapted to clarify this.

Filled circles: data conforms to model-aided selection criteria and was used for fitting. Unfilled circles: data does not conform to model-aided selection criteria. X: data did not pass quality control criteria and was not considered for model-aided analysis.

5. Fig S4D: “Dashed lines represent the 95% confidence interval...” means the gray dashed lines? The next sentence refers to the red dashed lines.

Yes indeed, sorry for the confusion and thank you for pointing it out. Gray dashed horizontal and vertical lines represent the 95% confidence interval of the regression. This has been added to the legend.

6. Fig S5: The “A” panel notation is cut off. Should it be removed since there are no other panel elements?

Thank you for pointing out this error - it has been corrected.

7. Fig S7B: Typo of “Confidence Interval” in subplot title.

Thank you for pointing out this error - it has been corrected.

Reviewer #4 (Remarks to the Author):

Lefaudeux et al. propose a mechanistic model of mRNA transport from chromatin to nucleoplasm to cytoplasm for immune response genes in mouse. The analysis is based on very interesting RNA-seq data measuring mRNA quantity at quite high temporal resolution in the three compartments. The fitted kinetic model enables the estimation of several rates related to the appearance, disappearance and transport of mRNA from one compartment to another, for each gene individually. Results show that transport rates from chromatin to cytoplasm greatly vary

from one gene to another, and seem to be essentially linked to gene length and the number of introns, but not to the chromatin states (histone marks) of the genes.

The question addressed in this paper and the data produced are very interesting IMO. The kinetic model is sound and the methodology developed to assess the model and conclusions make sense. We're pleased with the reviewer's appreciation for the importance of the question and the quality of the work. We also appreciate that the reviewer is not familiar with dynamical systems models consisting of a set of differential equations. This approach is not unusual in different branches of biology in which dynamic phenomena take place, including ecology, cell biology, signaling, immune cell dynamics.

However, as a statistical machine learning researcher, I didn't succeed to completely understand the mathematics developed to estimate the parameters of the differential equations. From what I understand, the methodology used in the paper may be quite usual for the physicist community, but it is quite different from what is done in the ML literature. As a consequence, I don't know if this is really helpful, but here are some points I have raised:

- in the ML literature, one of the first assessment (and certainly the most important) that is done when a model has been trained is to estimate the error of the model on an independent test set, i.e. on a set of examples that have not been used to estimate the parameters of the model. Very surprisingly I didn't see this important verification here. The authors provide some comparison between real data and simulated data, and even provide a measure of fit quality, but from what I understand these comparisons were done directly with the data used to estimate the parameters of the model. I suppose that such a test on independent data could also be done here, for example on some time points left out before training (maybe using a cross-validation procedure), or using the data of another replicate.

We appreciate this comment, and completely agree that an ML model (which is not anchored in knowledge of molecular mechanism) cannot be used to learn about the mechanism, but is solely used for prediction. Its predictive power must be assessed with an independent dataset. While the thrust of the paper uses an ODE model (which represents molecular mechanisms), our attempts to determine whether chromatin features are predictive of the derived rate constants use an ML model (Fig.5).

For the ODE modeling portion of the work, we fit the model to 2 replicates independently and compared the results of the parameters obtained: the results were very similar which strengthened the conclusions. We also compared the mRNA half-lives obtained from the model to the values obtained with an alternate method and different dataset (Fig.3E).

For the ML portion of the work in Fig.5, we did indeed split the data into independent training and testing sets. The training was done with cross-validation procedure and we find little predictive power on the independent testing dataset.

- concerning the fit quality statistics, I don't understand the sentence "As the negative log likelihood is not easily compared between genes given that it also depends on the expression level". Please explain what do you mean, and why you cannot use directly the cost function used for training as a fit quality index.

We used the negative log likelihood for parameter fitting. "Fit quality" represents what would be perceived by as good fit. The major difference is that it takes into account the autocorrelation of

the residues (differences between data and model). If the fits of two different genes had the same likelihood but one was consistently below the data values it would be perceived as a worse fit than another gene that is sometimes below, sometimes above the data, especially if the timecourse data for the second gene are more jagged. An abbreviated version of this explanation has been included in the Methods.

- In part Method, the section "Error model and cost function" is not clear. I don't understand the logic between the different equations, and the final expression of the likelihood is cryptic for me. As the final goal is a regression problem, I don't understand why the cost function does not involve a term related to the classical mean square error. Also, please number the formulae and refer to these numbers in the text.

The Methods section on the error and cost function has been revised, split and expanded. The error model part has been rewritten to make it clearer with more details.

The usual mean square error assumes a normal distribution of the error which is not what is observed for RNAseq, and read counts are often considered to be distributed following a negative binomial distribution (for example by edgeR, DESeq, cuffdiff from cufflinks...). Moreover, to be able to fit individual replicates we need to include in this error model a term that corresponds to temporal error (as for dynamically changing analytes an error in the actual timepoint of sample collection can have an impact on the observed measurement). Hence, we developed an error model to obtain the negative binomial distribution that includes a temporal error term.

- At several places in the draft the authors mention "the likelihood of the data". This doesn't make sense for me: the data is given, so it cannot be "likely". The likelihood refers to the model, not to the data.

Indeed, we apologize for this poor wording. This was meant to describe the likelihood of reproducing the data given the model and a parameter set. We have corrected the text.

- In the RBP motif analysis, the reported p-values are quite high (not very good). Did they have been corrected for multiple testing problem? Also, please provide additional information: which motif database has been used exactly? Which were the parameters of AME? etc.

The parameters and databases used have been added more precisely to the methods. Indeed, the p-values given are adjusted with bonferroni correction.

- For some figures the resolution is very bad and the text inside these figures is almost unreadable.

We apologize for this, which seems to have been the result of embedding the figures in a single document. In the revised submission we kept them separate.

Reviewer #5 (Remarks to the Author):

The paper explores mRNA transport and degradation for genes associated to the innate immune response. The overall analysis considers 3 subsequent states of each mRNA, from chromatin, to nuclear envelope to cytoplasmic mRNA for finally to be degraded. The analysis is thorough and

contains useful quantitative lessons on typical rates of mRNA in mammalian cells.

I recommend publication, with some clarifications:

1) Rate parameters are presumably in 1/minutes? (couldn't find that in paper)

Thank you for pointing out this omission. Yes, rates are in 1/min. A table has been added with a description and units of all parameter and combination of parameters in the revised paper.

2) Fig. 3 D. maybe explain why k_1/k_2 is interesting? (i.e. indicate how much there is on npRNA to cytoRNA)

We do not mean to suggest that this ratio is biologically meaningful. We mention this ratio as an example for the phenomenon that even when individual parameters are not identifiable their combination may still be.

3) Fig. 4A and B please use same x axis for easier comparison. Maybe even put panel E below these panels with same x-axis to allow easier comparisons.

Panel A describes the Export efficiency and B the effective transport rates. They have different units, and so the values are not really comparable. However, panel E shows the same quantity as panel B, and so the axis has now been adjusted to be the same (from -2.5 to +0.5) and panel D and E have been switched.

4) Fig 7B) indicate that k_{deg} is about a factor 1 to 10 slower than k_2 , indicating that degradation typically is slower than last step in transport. However, it could still be faster than whole transport (effective transport rate).

We thank the reviewer for this observation. The fact that k_{cyto_deg} is generally slower than k_2 is indeed a reason why k_{cyto_deg} plays a more important role in determining responsiveness.

5) The whole discussion associated to fig 7 seems equivalent to the old fact that if x is produced according to $dx/dt = a - x \cdot k_{deg}$ then k_{deg} sets the time to reach half max while A/k_{deg} sets the max level. Thus it is not surprising that k_{deg} sets responsiveness (Fig. 7C trivial)

We agree with the reviewer that the left panel of 7C is trivial, it functions as a 'control' or 'sanity check' for the right panel. Also, we feel for the broader experimental biologists it is useful to provide this background. Indeed, Reviewer 3 is raising a question about it. Hence, we would prefer to keep this figure in the manuscript.

6) In Fig. 7D,E,F) what is relative transport? Another name for effective transport? Panel D is interesting, but please explain better, perhaps using $dx/dt = a - x \cdot k_{deg}$ with $A = \text{transport}$ roughly scaling with k_{deg} .

We apologize for the confusion. It should be labelled effective transport. This has been corrected. Panel D shows that there is a correlation between effective transport rate and cytoplasmic degradation rate. The correlation has indeed a linear form supporting the idea that the two processes are coordinated, either mechanistically linked or co-evolved. We have tried to provide a fuller description.

Overall, what about total level of mRNA in first compartment (RNAc₁)? Eg. does total mRNA level correlate with any of the quantities in Fig. 7D? Or does total level of final produced protein in cytoplasm?

We thank the reviewer for this suggestion. We examined the correlation between peak cyto mRNA expression with effective transport rates and cytoplasmic mRNA half-life (Fig. 7F). The correlations are poor, indicating that mRNA half-life is not a determinant of expression level, as a priori consideration may suggest. The anticorrelation with effective transport rates ensures that mRNA abundance can be regulated independently of mRNA half-life.

REVIEWERS' COMMENTS

Reviewer #1 (Remarks to the Author):

The authors have done a somewhat decent job in addressing this reviewers' concerns.

However, some points require further clarification:

It is disappointing that the authors did not perform 4sU experiments as requested. Instead, they mention in passing that pilot studies showed that it is more reliable n 4sU labeling method. However they don't expand on this or show the pilot data that demonstrates why this is the case. It would be useful to see this data.

They also mention that caRNA measurements indeed show a major reduction in nascent mRNA synthesis in the LPA-tolerized conditions, but again don't show the data (or don't refer to which figure it is in).

Lastly, Figure 7 is confusing. Figure 7F isn't referred to either in the figure legend or in the text but is in the figure.

Reviewer #2 (Remarks to the Author):

The authors have addressed most of this reviewer's points.

However, regarding the previous point #6, the Methods still seem to suggest that, depending on the histone mark, a set of between 4-8 windows with a width of 2,500-8,333 bp each and centered around the TSS of each gene was used to average ChIP signals in each window. This is in contrast to the authors' response that the sum of all reads associated with the gene were used. This should still be clarified.

Reviewer #3 (Remarks to the Author):

The authors have addressed all concerns and then manuscript can be published.

Reviewer #4 (Remarks to the Author):

The authors have responded to my main concerns. I do not agree completely with one of the answers but this is related to a methodological issue that I feel is beyond the scope of the paper. So in my opinion the paper can be published like this. I provide below some arguments about these points for scientific purposes only.

1) "We appreciate this comment, and completely agree that an ML model (which is not anchored in knowledge of molecular mechanism) cannot be used to learn about the mechanism, but is solely used for prediction. Its predictive power must be assessed with an independent dataset. (...) For the ODE modeling portion of the work, we fit the model to 2 replicates independently and compared the results of the parameters obtained: the results were very similar which strengthened the conclusions. We also compared the mRNA half-lives obtained from the model to the values obtained with an alternate method and different dataset (Fig.3E)."

This does not completely respond to my question. I still believe that mechanistic models could be

assessed on independent data as it is done for statistical models, and I don't understand why this important step could be skipped just because models are ODE. The purpose of this assessment is to check for issues related to over-fitting, and over-fitting can also occur for mechanistic models. The fact that 2 models fitted on 2 independent data have very similar parameter values is indeed encouraging, but "very similar" is a quite subjective notion that is not as convincing as measuring the error on independent data.

2) Q: "At several places in the draft the authors mention "the likelihood of the data". This doesn't make sense for me: the data is given, so it cannot be "likely". The likelihood refers to the model, not to the data."

A: "Indeed, we apologize for this poor wording. This was meant to describe the likelihood of reproducing the data given the model and a parameter set. We have corrected the text."

I do not completely agree with this answer. The likelihood readily refers to the model, and is defined as the probability (density) of observing the data given the model. In statistics, it is often denoted as $L(M:D) = P(D|M)$. So, the right way to use the word "likelihood" is either just "likelihood" without anything else (which is the most usual way in the literature) or "model likelihood".

Reviewer #5 (Remarks to the Author):

I am happy with the corrected manuscript and recommend publication without further delay.

REVIEWERS' COMMENTS

Reviewer #1 (Remarks to the Author):

The authors have done a somewhat decent job in addressing this reviewers' concerns. We thank the reviewer for evaluating our revised manuscript.

However, some points require further clarification:

It is disappointing that the authors did not perform 4sU experiments as requested. Instead, they mention in passing that pilot studies showed that it is more reliable than 4sU labeling method. However, they don't expand on this or show the pilot data that demonstrates why this is the case. It would be useful to see this data.

Our understanding is that 4sU labeling protocols can be used to quantify nascent RNA synthesis, and may also be used ("pulse-chase") for quantifying mRNA half-life. In our experience the limitations of 4sU are sensitivity (must effectively flood the endogenous pool of nucleotides with the analog) and perturbing cell homeostasis and cell toxicity.

When the transcriptome is in steady state, labeling with 4sU is the best available approach (despite its shortcomings) to quantify synthesis rates. In this project, we focused on LPS-inducible genes so that chromatin-associated RNA can be sequenced directly without the need for labeling and the associated technical concerns. This approach was used by the Doug Black and Steve Smale labs previously to gain insights about the coordination of pre-mRNA processing, transcription, and transport across chromatin, nucleoplasm, and cytoplasm (Cell. 2012 Jul 20;150(2):279-90, RNA. 2013 Jun;19(6):811-27). Based on these optimized protocols, our study quantifies these processes for 212 LPS-inducible genes for which the data is of sufficient quality. Aside from transcriptional initiation rates, elongation and intron removal rates could also be quantified from this caRNAseq data.

For the application of measuring mRNA half-life we did compare 4sU and ActD protocols. In our experience, both are fraught with difficulties and artefact. In our experience, labelling efficiency was low for long-lived mRNAs such that their half-lives could not be quantified. In the case of ActD, even if the quantitation resulted in underestimates (compared to model inferred half-lives, Fig. 3E), one could at least state that the half-life was >8hrs (with 4sU there is simply an absence of data due to insufficient signal).

They also mention that caRNA measurements indeed show a major reduction in nascent mRNA synthesis in the LPA-tolerized conditions, but again don't show the data (or don't refer to which figure it is in). We apologize for not pointing out the Figure panel: it is Figure 6A and B.

Lastly, Figure 7 is confusing. Figure 7F isn't referred to either in the figure legend or in the text but is in the figure. We thank the Reviewer for pointing this out: this panel was added in response to a suggestion by Reviewer 5 (who thought the work in Figure 7 was particularly interesting). We apologize for omitting a figure legend. We have corrected this.

Reviewer #2 (Remarks to the Author):

The authors have addressed most of this reviewer's points.

However, regarding the previous point #6, the Methods still seem to suggest that, depending on the histone mark, a set of between 4-8 windows with a width of 2,500-8,333 bp each and centered around the TSS of each gene was used to average ChIP signals in each window. This is in contrast to the authors' response that the sum of all reads associated with the gene were used. This should still be clarified.

The Methods description includes a variety of analyses that were done, only a subset is shown in the main Figure. The Legend of the main Figure states. "D. Sum of ChIP-seq signals of indicated histone mark associated with the gene do not show a correlation with the effective transport rate. (Alternatively, window sizes of 1kb or 5kb along the gene were tried but yielded no better correlation.)". We have made further edits to the Methods description to avoid confusion.

Reviewer #3 (Remarks to the Author):

The authors have addressed all concerns and then manuscript can be published.

Reviewer #4 (Remarks to the Author):

The authors have responded to my main concerns. I do not agree completely with one of the answers but this is related to a methodological issue that I feel is beyond the scope of the paper. So in my opinion the paper can be published like this. I provide below some arguments about these points for scientific purposes only.

We thank the reviewer for their appreciation of the work and we value the additional points and discussion below.

1) "We appreciate this comment, and completely agree that an ML model (which is not anchored in knowledge of molecular mechanism) cannot be used to learn about the mechanism, but is solely used for prediction. Its predictive power must be assessed with an independent dataset. (...) For the ODE modeling portion of the work, we fit the model to 2 replicates independently and compared the results of the parameters obtained: the results were very similar which strengthened the conclusions. We also compared the mRNA half-lives obtained from the model to the values obtained with an alternate method and different dataset (Fig.3E)."

This does not completely respond to my question. I still believe that mechanistic models could be assessed on independent data as it is done for statistical models, and I don't understand why this important step could be skipped just because models are ODE. The purpose of this assessment is to check for issues related to over-fitting, and over-fitting can also occur for mechanistic models. The fact that 2 models fitted on 2 independent data have very similar parameter values is indeed encouraging, but "very similar" is a quite subjective notion that is not as convincing as measuring the error on independent data.

We agree with the reviewer that mechanistic models run the risk of being overfit. Throughout our study we report the range of parameter values that fit the data. Indeed, that reveals that some parameters could never be meaningfully constrained by the data, and some parameters could not be meaningfully fit for some genes, or in some conditions (tolerized in Fig. 6). We fit the model to 2 independent experiments, and we report the quantitative comparison of the results (e.g. all panels in Figure 2). How well the results correspond depends on the gene of course, but for each there is a quantitative evaluation. When our response says that the fits are "very similar" that is simply the highest level summary, but we submit that the quantitative evaluation of the replicate data is rigorous.

2) Q: "At several places in the draft the authors mention "the likelihood of the data". This doesn't make sense for me: the data is given, so it cannot be "likely". The likelihood refers to the model, not to the data."

A: "Indeed, we apologize for this poor wording. This was meant to describe the likelihood of reproducing the data given the model and a parameter set. We have corrected the text."

I do not completely agree with this answer. The likelihood readily refers to the model, and is defined as the probability (density) of observing the data given the model. In statistics, it is often denoted as $L(M:D) = P(D|M)$. So, the right way to use the word "likelihood" is either just "likelihood" without anything else (which is the most usual way in the literature) or "model likelihood".

We agree with the reviewer, and we do not quite follow what the distinction is. Based on the definition $L(M:D) = P(D|M)$, likelihood measures the probability (density) of observing the data given the model. That data is the result of simulations using the model parameters. The simulated data is compared to the experimental data. Hence we write: "the likelihood of reproducing the [experimental] data given the model and a parameter set". We hope that is acceptable.

Reviewer #5 (Remarks to the Author):

I am happy with the corrected manuscript and recommend publication without further delay.